# POTENTIAL OUTCOME IMPUTATION
# FOR CATE ESTIMATION

## ABSTRACT

One of the most significant challenges in Conditional Average Treatment Effect (CATE) estimation is the statistical discrepancy between distinct treatment groups. To address this, we propose a model-agnostic data augmentation method for CATE estimation. We first derive regret bounds for general data augmentation methods, indicating that reduced group discrepancy and low imputation error enhance CATE estimation. Inspired by this, we introduce a contrastive learning approach that reliably imputes missing potential outcomes for a selected subset of individuals based on a similarity measure. These reliable imputations augment the original dataset, reducing the discrepancy between treatment groups while inducing minimal imputation error. The augmented dataset can then be used to train standard CATE estimation models. We provide theoretical guarantees and extensive numerical studies, demonstrating our approach's effectiveness in improving the accuracy and robustness of various CATE estimation models.

## 1 INTRODUCTION

One of the most significant challenges for Conditional Average Treatment Effect (CATE) estimation is the statistical discrepancy between distinct treatment groups (Goldsmith-Pinkham et al., 2022). While Randomized Controlled Trials (RCTs) mitigate this issue (Rubin, 1974; Imbens & Rubin, 2015), they can be expensive, unethical, and unfeasible to conduct. Consequently, we are often constrained to rely on observational studies, which are susceptible to the aforementioned issue. To address this, we introduce a **model-agnostic data augmentation method**, comprising two key steps. First, our approach **identifies a subset of individuals** whose counterfactual outcomes can be reliably imputed. Subsequently, it **performs imputation for the missing counterfactual outcomes** of these selected individuals, thereby augmenting the original dataset with these imputed values. See Figure 1a for a visual illustration of the pipeline. Importantly, our method functions as a data pre-processing module that remains agnostic to the choice of the subsequent model employed for CATE estimation.

**Motivation.** Our method is motivated by an observed **trade-off** between **(i)** the statistical discrepancy across treatment groups and **(ii)** the error in counterfactual outcome imputation. Consider the scenario with a binary treatment assignment. In this context, no individual can appear in both the control and treatment groups due to the inaccessibility of counterfactual outcomes (Holland, 1986). Suppose that, with the sole aim of reducing discrepancies across treatment groups, we **randomly impute** the missing counterfactual outcomes and then integrate each individual, along with their randomly imputed outcomes, into the original dataset. This procedure ensures that the control and treatment groups have **identical individuals**, effectively **eliminating all discrepancies**. However, it is obvious that any model trained on such a randomly augmented dataset would exhibit poor performance due to the substantial errors introduced by the random imputation. This trade-off is illustrated in Figure 1b where *increasing* level of data augmentation simultaneously *decreases* the discrepancy across treatment groups and *increases* the imputation error. Motivated by this, our approach aims to address this challenge by identifying *a subset of individuals* for whom the counterfactual outcomes can be **reliably imputed**. We formalize this idea with a generalization bound in Section 4 which affirms an intuitive conclusion that *an augmentation method with low counterfactual outcome imputation error can enhance CATE estimation*.

**Algorithm.** To this end, our approach utilizes contrastive learning to identify the individuals whose counterfactual outcomes can be reliably imputed. Specifically, it learns a representation space and

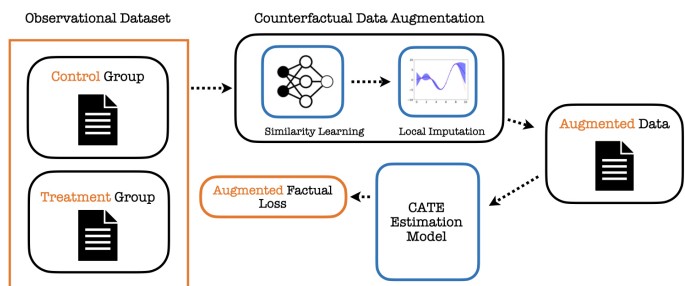
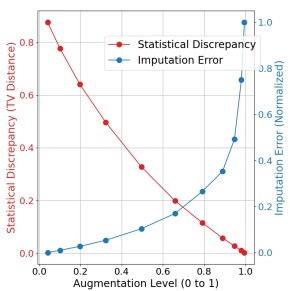

(a) Similarity learning is used to select a subset of individuals, followed by reliable local imputations to generate their counterfactuals. These imputations augment the original dataset, reducing the statistical discrepancy between treatment groups while minimizing imputation error. The augmented data is then used to train off-the-shelf CATE estimation models, improving their accuracy and robustness.

(b) Trade-off between statistical discrepancy and imputation error across different augmentation levels (0 to 1). A full description of the synthetic toy dataset and implementation details can be found in Appendix D.2.

Figure 1: (a) Overview of the proposed model-agnostic data augmentation method for CATE estimation, and (b) the observed trade-off that motivated the proposed method.

a similarity measure such that within this learned representation space, *close* individuals identified by the similarity measure exhibit *similar* potential outcomes. This *smoothness* property guarantees *reliable counterfactual outcome imputation through local approximation* **for individuals with a sufficient number of close neighbors from the alternative treatment group**. After identifying these individuals, we impute their counterfactual outcomes by utilizing the factual outcomes of their proximate neighbors (from the alternative treatment group). Importantly, the smoothness property, which results from contrastive learning, ensures that the imputation can be achieved locally with simple models that require minimal tuning. We explore two distinct methods for imputation: *linear regression* and *Gaussian Processes*.

**Theoretical and Empirical Validation.** To comprehensively assess the efficacy of our data augmentation technique,

- we theoretically establish that ***our approach asymptotically generates datasets whose probability densities converge to those of RCTs***;
- we provide ***non-asymptotic generalization bounds*** for the performance of CATE estimation models trained with our augmented data;
- our empirical results further demonstrate the efficacy of our method, showcasing ***consistent enhancements in the performance of various CATE estimation models***, including TARNet, CFR-Wass, and CFR-MMD (Shalit et al., 2017), S-Learner and T-Learner integrated with neural networks, Bayesian Additive Regression Trees (BART) (Hill, 2011; Chipman et al., 2010; Hill et al., 2020) with X-Learner (Künzel et al., 2019), and Causal Forests (CF) (Athey & Imbens, 2016) with X-Learner.

## 2 RELATED WORKS

One of the fundamental tasks in causal inference is to estimate *Average Treatment Effects* (ATE) and *Conditional Average Treatment Effects* (CATE) (Neyman, 1923; Rubin, 2005). Various methods have been proposed for ATE estimation, including Covariate Adjustment (Rubin, 1978), Propensity Scores (Rosenbaum & Rubin, 1983), Doubly Robust estimators (Funk et al., 2011), Inverse Probability Weighting (Hirano et al., 2003), and recently Reisznet (Chernozhukov et al., 2022). While these methods are successful for ATE estimation, they are not directly applicable to CATE estimation.

On the other hand, recent advances in machine learning have led to new approaches for CATE estimation, such as decision trees (Athey & Imbens, 2016), Gaussian Processes (Alaa & Van Der Schaar, 2017), Multi-task deep learning ensemble (Jiang et al., 2023), Generative Modeling (Yoon et al., 2018), and representation learning with deep neural networks (Shalit et al., 2017; Johansson et al.,

2016). It is worth noting that alternative approaches for investigating causal relationships exist, such as do-calculus, proposed by Pearl (Pearl, 2009a;b). Here, we adopt the Neyman-Rubin framework. At its core, the CATE estimation problem can be seen as a missing data problem (Rubin, 1974; Holland, 1986; Ding & Li, 2018) due to the unavailability of the counterfactual outcomes. In this context, we propose a new data augmentation approach for CATE estimation by imputing certain missing counterfactuals. Data augmentation, a well-established technique in machine learning, serves to enhance model performance by artificially expanding the size of the training dataset (Van Dyk & Meng, 2001; Chawla et al., 2002; Han et al., 2005; Jiang et al., 2020; Chen et al., 2020a; Liu et al., 2020; Feng et al., 2021).

A crucial aspect of our methodology is the identification of similar individuals. There are various methods to achieve this goal, including propensity score matching (Rosenbaum & Rubin, 1983), Mahalanobis distance matching (Imai et al., 2008), and nearest neighbor matching algorithms (Holzmann & Meister, 2024; Lin et al., 2023). Nonetheless, these methods pose significant challenges, particularly in scenarios with large sample sizes or high-dimensional data, where they suffer from the curse of dimensionality. Recently, Perfect Match (Schwab et al., 2018) is proposed to leverage importance sampling to generate replicas of individuals. It relies on propensity scores and other feature space metrics to balance the distribution between the treatment and control groups during the training process. In contrast, we utilize contrastive learning to construct a similarity metric within a representation space. Our method focuses on imputing missing counterfactual outcomes for a selected subset of individuals, without creating duplicates of the original data points. While the Perfect Match method is a universal CATE estimator, our method is a model-agnostic data augmentation method that serves as a data preprocessing step for other CATE estimation models.

## 3 PRELIMINARIES

Let $T \in \{0, 1\}$ be a binary treatment assignment, $X \in \mathcal{X} \subset \mathbb{R}^d$ be the covariates (features), and $Y \in \mathcal{Y} \subset \mathbb{R}$ be the factual (observed) outcome. For each $j \in \{0, 1\}$, we define $Y_j$ as the *potential outcome* (Rubin, 1974), which represents the outcome that would have been observed if only the treatment $T = j$ was administered. The random tuple $(X, T, Y)$ jointly follows the *factual (observational) distribution* denoted by $p_F(x, t, y)$. Let $D_F = \{(x_i, t_i, y_i)\}_{i=1}^n$ denote a dataset that consists of $n$ observations independently sampled from $p_F$ where $n$ is the number of observations.

**Definition 3.1** (CATE). The Conditional Average Treatment Effect (CATE) is defined as:

$$\tau(x) = \mathbb{E}[Y_1 - Y_0 | X = x]. \tag{1}$$

Throughout this work, we make the standard assumptions of **positivity**, i.e., $0 < p_F(T = 1|X) < 1$, and **conditional unconfoundedness**, i.e., $(Y_1, Y_0) \perp\!\!\!\perp T|X$, so that CATE is identifiable (Robins, 1986; Imbens & Rubin, 2015). Let $\hat{\tau}(x) = h(x, 1) - h(x, 0)$ denote an estimator for CATE where $h$ is a hypothesis $h : \mathcal{X} \times \{0, 1\} \to \mathcal{Y}$ that estimates the underlying causal relationship $f$ between $(X, T)$ and $Y$.

**Definition 3.2** (PEHE). The Expected Precision in Estimating Heterogeneous Treatment Effect (PEHE) (Hill, 2011) is defined as:

$$\varepsilon_{\text{PEHE}}(h) = \int_{\mathcal{X}} (\hat{\tau}(x) - \tau(x))^2 p_F(x) dx \tag{2}$$

$\varepsilon_{\text{PEHE}}$ is widely used as the performance metric for CATE estimation. However, directly estimating $\varepsilon_{\text{PEHE}}$ from observational data $D_F$ is a non-trivial task, as it requires knowledge of the counterfactual outcomes. This challenge underscores that models for CATE estimation need to be robust to overfitting the factual distribution. ***Notably, our empirical results (in Section 7) indicate that our method mitigates the risk of overfitting for various CATE estimation models***. Apart from $\varepsilon_{\text{PEHE}}$, we will also consider the following loss function in our theoretical results.

**Definition 3.3.** For a distribution $p$ over $(X, T, Y)$ and a hypothesis $h$, the loss function $\mathcal{L}_p(h)$ is defined as:

$$\mathcal{L}_p(h) = \int (y - h(x, t))^2 p(x, t, y) \, dx \, dt \, dy,$$

## 4 UNDERSTANDING DATA AUGMENTATION FOR CATE ESTIMATION

We first present a generalization bound for the performance of CATE estimation models *trained using an augmented dataset*. This result serves as the ***theoretical foundation*** of our proposed augmentation method in Section 5.

Given the factual dataset $D_F$ with $n$ samples, a data augmentation algorithm based on counterfactual imputation has two main components:

- *Component I:* identifying a subset $\mathcal{R}_n \subset \mathcal{X} \times \{0, 1\}$, where $\mathcal{R}_n^t \subset \mathcal{X}$ for $t \in \{0, 1\}$ is the projection for the treatment and control groups on which to perform data augmentation.
- *Component II:* imputing the missing potential outcomes for individuals in $\mathcal{R}_n$ with an algorithm $\tilde{f}_n : \mathcal{R}_n \to \mathcal{Y}$.

**Notations.** Let $p_{AF}(x, t, y)$ be the distribution of $(X, T, Y)$ in the augmented dataset. Due to space limitation, we defer the mathematical definition of $p_{AF}(x, t, y)$ to Appendix C.1. Let $p_{AF}(x, t)$ and $p_{RCT}(x, t)$ represent the marginal distributions of $(X, T)$ when sampled from the augmented dataset and RCTs, respectively. To establish the generalization bound, we assume that there is a true potential outcome function $f$ such that $Y = f(X, T) + \eta$ with $\eta$ verifying that $\mathbb{E}[\eta] = 0$. Let $\beta \in (0, 1)$ denote the percentage of the total data pointed selected for counterfactual imputation, i.e., $\beta = n_a/n$ where $n_a$ is the number of points selected for imputation and $n$ is the total number of samples in the dataset.

**Theorem 4.1** (Generalization Bound). *Let $h$ be a hypothesis, its $\varepsilon_{PEHE}$ is upper bounded as follows:*

$$\varepsilon_{PEHE}(h) \leq 4 \cdot \left( \underbrace{\mathcal{L}_{p_{AF}}(h)}_{(I)} + 2 \underbrace{V\big(p_{RCT}(X, T), p_{AF}(X, T)\big)}_{(II)} + \underbrace{\frac{\beta}{1 + \beta} \cdot b_{\mathcal{A}}(n),}_{(III)} \right) \quad (3)$$

*where $V(g_1, g_2) = \frac{1}{2} \int_{\mathcal{S}} |g_1(s) - g_2(s)| ds$ is the total variation distance between two densities, and*

$$b_{\mathcal{A}}(n) = \mathbb{E}_{X, T \sim q}\big[\|f(X, T) - \tilde{f}_n(X, T)\|^2\big],$$

*where $q(x, 1 - t) = \frac{p_F(x, 1-t)}{\alpha} \mathbb{1}_{\mathcal{R}_n}$ with $\alpha = \int p_F(x, 1 - t) \mathbb{1}_{\mathcal{R}_n}(x, 1 - t) dx dt$.*

*Remark* 4.2. We note that term *(I)* in Theorem 4.1 is essentially the training loss of a hypothesis $h$ on the augmented dataset while ***term (II) characterizes the statistical similarity*** between the individuals' features in the augmented dataset and those generated from an RCT. Meanwhile, ***term (III) characterizes the accuracy of the data augmentation method***.

Hence, this theorem highlights the trade-off between the disparity across treatment groups and the imputation error, which is empirically illustrated in Figure 1b. More importantly, it underscores that ***simultaneously minimizing (i) the statistical disparity across treatment groups and (ii) the imputation error can enhance the performance of CATE estimation models***. Thus, we reach a quite intuitive conclusion: *an augmentation method with low counterfactual imputation error can help CATE estimation*. It is also essential to highlight that if the local regression module can achieve more accurate estimation with more samples (e.g., local Gaussian Process) $b_{\mathcal{A}}(n)$ will converge to 0, as proved in Section 6.

## 5 CONTRASTIVE COUNTERFACTUAL AUGMENTATION

Motivated by Theorem (4.1) and as discussed in the introduction, the effectiveness of counterfactual augmentation depends on reliable imputation. To this end, we propose to learn a representation space along with a similarity measure such that: within this representation space, individuals classied as similar by the similarity measure should exhibit similar potential outcomes. In other words, an individual's potential outcome exhibit a strong correlation with those of its nearby neighbors. *This smoothness property ensures reliable imputation through local approximation*. As a result, for the individuals who possess a sufficient number of close neighbors from the alternative treatment group, we can reliably impute their *counterfactual outcomes* using the *factual outcomes* of their nearby neighbors and, as established above, ***augmenting the original dataset with these reliable imputations can enhance CATE estimation***.

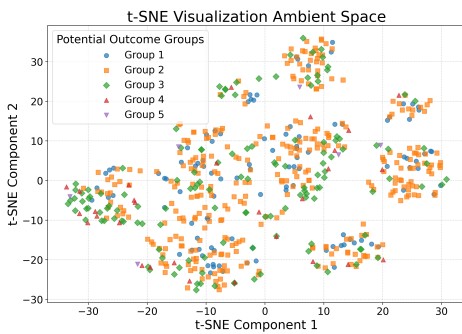 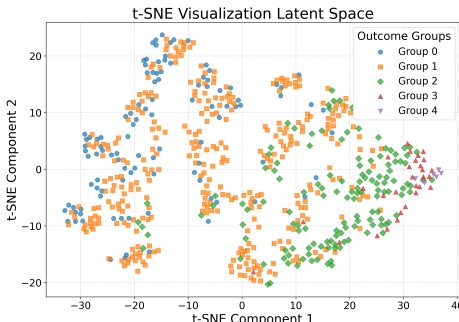

Figure 2: t-SNE visualization of IHDP features and potential outcome $Y_0$ in the ambient space (left) and the latent space (right) learned by contrastive learning. Groups are defined by dividing the potential outcome $Y_0$ values into five equal intervals from smallest to largest, with each individual labeled based on the value of its potential outcome.

**Overview.** We propose COntrastive COunterfactual Augmentation (COCOA) with two components. The ***first component*** is a classifier $g_\theta$, trained with contrastive learning (Le-Khac et al., 2020; Jaiswal et al., 2020) to learn a representation space and a similarity measure. For a given individual $x$, $g_\theta$ identifies $x$'s close neighbors, that is, individuals in the dataset $D_F$ who are likely to exhibit similar outcomes when subjected to the same treatment assignment as $x$. The ***second component*** is a local regressor $\psi$, which imputes the counterfactual outcome for $x$ after being fitted to its close neighbors.

Specifically, for $t \in \{0, 1\}$, we use $D^t \subset D_F$ to denote the factual observations in treatment group $t$, i.e, $D^t = \{(x_i, t_i, y_i) \in D_F | t_i = t\}$. The counterfactual imputation has the following steps:

1. ***Neighbor Identification.*** For a given individual $x$ within treatment group $t$ whose counterfactual outcome (that is, potential outcome under treatment $1 - t$) needs to be imputed, the trained classifier $g_\theta$ first identifies a set of close neighbors to $x$, denoted by $D_x \subset D^{1-t}$. In particular, $D_x$ are individuals in treatment group $1 - t$ who are likely to have similar potential outcomes to $x$ under treatment $1 - t$.

2. ***Local Approximation.*** Subsequently, the non-parametric regressor $\psi$ utilizes the factual outcomes in $D_x$ to estimate the counterfactual outcome of $x$: $\widehat{y}_x = \psi(x, D_x)$.

3. ***Augmentation.*** Finally, the imputed outcome of $x$ is incorporated into the dataset, i.e., $D_A = D_A \cup \{(x, 1-t, \widehat{y}_x)\}$ where $D_A$ is initialized as $D_A = D_F$.

**Selective Imputation.** As discussed in Section 1 and shown by Theorem (4.1), minimal counterfactual imputation error plays a crucial role in the success of data augmentation. ***To ensure the reliability of these imputations, we only perform imputations for individuals who possess a sufficient number of close neighbors***. Thus, we only estimate the counterfactual outcome of $x$ if $|D_x| \geq k$, where $k$ is a pre-determined parameter that controls estimation accuracy. In the worst case, no individuals will meet the imputation criteria, resulting in *no augmentation* of the dataset. It is important to note that unlike standard CATE models, COCOA does not generalize to unseen samples. Its goal is to identify individuals within the dataset and impute their counterfactual outcomes, thereby augmenting the dataset to improve CATE models' predictions on unseen samples. *The augmented dataset $D_A$ is then used as the training dataset for CATE estimation models.* See Algorithm 1 for pseudocode of COCOA. We next discuss the classifier $g_\theta$ and the regressor $\psi$ in detail.

**Contrastive Learning Module.** Contrastive (representation) learning methods (Wu et al., 2018; Bojanowski & Joulin, 2017; Dosovitskiy et al., 2014; Caron et al., 2020; He et al., 2020; Chen et al., 2020b; Trinh et al., 2019; Misra & Maaten, 2020; Tian et al., 2020) are based on the principle that similar individuals should be associated with closely related representations within an embedding space. This is achieved by training models to perform an auxiliary task: predicting whether two individuals are similar or dissimilar.

In the context of CATE estimation, we consider two individuals with *similar outcomes* under the same treatment as *similar individuals*. As individuals who are close in the original space may not generally verify this property, we utilize contrastive learning approaches to learn a space where this

---

**Algorithm 1** Contrastive Counterfactual Augmentation

---

**Input:** Factual dataset $D_F = \{(x_i, t_i, y_i)\}_{i=1}^n$; sensitivity parameter $\epsilon$; threshold $k$
**Output:** Augmented factual dataset $D_A$ as training dataset for CATE estimation models
Initialize $D_A = D_F$
Construct datasets $D_\epsilon^+$ and $D_\epsilon^-$ from $D_F$
Learn a parametric classifier $g_\theta$ with contrastive learning and $(D_\epsilon^+, D_\epsilon^-)$ by optimizing Equation 4.
**for** $i = 1$ **to** $n$ **do**
    Determine $x_i$'s close neighbors $D_{x_i} = \{(x_j, y_j)|j \in [n], t_j = 1 - t_i, g_\theta(x_i, x_j) = 1\}$
    **if** $|D_{x_i}| \geq k$ **then**
        Counterfactual Imputation $\widehat{y}_i = \psi(x_i, D_{x_i})$; Add $(x_i, 1 - t_i, \hat{y}_i)$ to $D_A$
    **end if**
**end for**

---

property holds. Figure 2 illustrates this: *with contrastive learning, the features of the individuals with similar potential outcomes are more clustered in the representation space, demonstrating the smoothness property that enables reliable local imputation.*

**Module Training.** The degree of similarity between outcomes is measured using a particular metric in the potential outcome space $\mathcal{Y}$. In our case, we employ the Euclidean norm in $\mathbb{R}^1$ for this purpose. With this perspective, given the factual (original) dataset $D_F = \{(x_i, t_i, y_i)\}_{i=1}^n$, we construct a ***positive dataset*** $D_\epsilon^+$ that includes pairs of similar individuals. Specifically, we define $D_\epsilon^+ = \{(x_i, x_j) : i, j \in [n], i \neq j, t_i = t_j, \|y_i - y_j\| \leq \epsilon\}$ where $\epsilon$ is user-defined sensitivity parameter specifying the desired level of precision. We also create a ***negative dataset*** $D^- = \{(x_i, x_j) : i, j \in [n], i \neq j, t_i = t_j, \|y_i - y_j\| > \epsilon\}$ containing pairs of individuals deemed dissimilar. Let $\ell : \{0, 1\} \times \{0, 1\} \to \mathbb{R}$ be any loss function for classification task. We learn a parametric classifier (neural network) $g_\theta : \mathcal{X} \times \mathcal{X} \to \{0, 1\}$ with parameter $\theta$ by optimizing the following objective function:

$$\min_\theta \sum_{(x,x') \in D_\epsilon^+} \ell(g_\theta(x, x'), 1) + \sum_{(x,x') \in D_\epsilon^-} \ell(g_\theta(x, x'), 0) \tag{4}$$

**Neighbor Identification.** For a given individual $x$ in $D_F$ within treatment group $t$, we utilize trained $g_\theta$ to identify its close neighbors $D_x \subset D_F$ for counterfactual imputation. Specifically, we iterate over all the individuals who received treatment $1 - t$ and employ $g_\theta$ to predict whether their potential outcomes are close to the potential outcome of $x$ under treatment $1 - t$. Hence, the selected neighbors of individual $x$[1] is defined as: $D_x = \{i \in [n] : t_i = 1 - t, g_\theta(x, x_i) = 1\}$. Note that we only impute the counterfactual outcome of $x$ if $|D_x| \geq k$ where $k$ is a pre-determined parameter to control the imputation error.

**Local Regression Module.** After identifying the nearest neighbors $D_x$, we employ a local regression module $\psi$ to impute the counterfactual outcomes. In this work, we explore two different types of local regression modules which are linear regression and Gaussian Process (GP). In experimental studies, we present results with GP using a Dot Product Kernel and defer the results for other kernels and linear regression to Appendix D.5. We opt for these straightforward function classes for local regression due to the following principles:

- *Local Approximation*: Complex functions can be locally estimated with simple functions, e.g., continuous functions and complex distributions can be approximated by a linear function (Rudin, 1953) and Gaussian distributions (Tjøstheim et al., 2021), respectively.

- ⋆ *Sample Efficiency*: If the class of the local linear regression module can estimate the true target function locally, then a class with less complexity will require fewer close neighbors for good approximations.

- † *Practicality*: A simpler class of $\psi$ requires less hyper-parameter tuning which is even more challenging in causal inference applications.

**Gaussian Process.** Gaussian Process (Seeger, 2004) offers robust solutions to regression problems. It is fully characterized by a mean function $m : \mathcal{X} \to \mathbb{R}$ and a kernel $K : \mathcal{X} \times \mathcal{X} \to \mathbb{R}_0^+$ and

---

[1]The terms "individual" and "indices of individuals" are used interchangeably.

it is denoted as $\mathcal{GP}(m, K)$. A GP is a random process $\phi(\mathcal{X})$ indexed by a set $\mathcal{X}$ such that any finite collection of these random variables follows a multivariate Gaussian distribution. Consider a finite index set of $n$ elements $\mathbf{x}_n \doteq \{x_i\}_{i=1}^n$, then the $n$-dimensional random variable $\phi(\mathbf{x}_n) \triangleq [\phi(x_1), \phi(x_2), \ldots, \phi(x_n)]$ follows a Gaussian distribution:

$$\phi(\mathbf{x}_n) \sim \mathcal{N}\big(m(\mathbf{x}_n), K(\mathbf{x}_n, \mathbf{x}_n)\big) \tag{5}$$

where $m(\mathbf{x}_n) = [m(x_1), \ldots, m(x_n)]$ is the mean and the $K(\mathbf{x}_n, \mathbf{x}_n)$ is a $n \times n$ covariance matrix whose element on the $i$-th row and $j$-th column is defined as $K(\mathbf{x}_n, \mathbf{x}_n)_{ij} \doteq K(x_i, x_j)$

**Potential Outcome Imputation.** Based on the principle of *Local Approximation*, if an individual $x$ in the factual dataset received treatment $t$, it is assumed that the potential outcome of $x$ under treatment $1 - t$ and those of its close neighbors (i.e., the individuals within $D_x$) follow a GP. Thus, after constructing $D_x$ using the method described above, the counterfactual outcome for $x$ is imputed as:

$$\widehat{y}_x^{1-t} = \psi(x, D_x) = \mathbb{E}[y^{1-t}|x, \{y_i\}_{i \in D_x}]. \tag{6}$$

Under the assumption of GP, $\widehat{y}_x^{1-t}$ has a closed-form solution. Let $\sigma(i)$ denote the $i$-th smallest index in $D_x$ and $K$ denote the kernel (covariance function) of GP. Then

$$\widehat{y}_x^{1-t} = \mathbf{K}_x^\top \mathbf{K}_{xx} \mathbf{y}, \tag{7}$$

where

$$\mathbf{K}_x = [K(x, x_{\sigma(1)}), \ldots, K(x, x_{\sigma(|D_x|)})], \quad \mathbf{y} = [y_{\sigma(1)}, \ldots, y_{\sigma(|D_x|)}]$$

and $\mathbf{K}_{xx}$ is a $|D_x| \times |D_x|$ matrix whose element on the $i$-th row and $j$-column is $K(x_{\sigma(i)}, x_{\sigma(j)})$. Finally, we append the tuple $(x, 1 - t, \widehat{y}_x^{1-t})$ into the factual dataset to augment the training data.

# 6 THEORETICAL INSIGHTS

This section explores the theoretical properties of COCOA, and aims to rigorously establish its efficacy. While the results presented here, similar to the extensive body of results on learning theory, are based on large-sample assumptions, they provide valuable insights into why local imputation methods such as COCOA are effective.

**Results Overview.** We present two main results:

- An asymptotic result showing that the augmented dataset distribution of COCOA converges to that of RCTs, thus ***effectively eliminating statistical disparity across treatment groups***
- A finite-sample regret guarantee for the GP local regressor showing that ***the imputation error can be provably controlled***.

These two results combined with Theorem 4.1 establish that *COCOA can be beneficial for CATE estimation*, which is also empirically verified later in Section 7.

**Notation.** We use $\mathcal{O}$ to denote the standard big-O notation for asymptotic behaviors and $\tilde{\mathcal{O}}$ to denote the big-O notation ignoring all the log terms. $||\cdot||_2$ denotes the Euclidean norm. For any two values $a, b \in \mathbb{R}$, we let $a \vee b = \max(a, b)$ and $a \wedge b = \min(a, b)$. Let $n_1$ and $n_0$ denote the number of individuals in the treatment and control groups, respectively. We define $u = \mathbb{P}(T = 1)$ as the probability of an individual being in the treatment group, and let $z = \frac{u}{1-u}$. Moreover, let

$$X^t \stackrel{d}{=} (X|T = t) \text{ and } \gamma = \mathbb{P}(\rho(X^1, X^0) \geq \epsilon) \in (0, 1),$$

where $\rho(\cdot, \cdot)$ denotes the distance metric between features (e.g. the contrastive learning distance) of the treatment and control groups, and $\epsilon$ is a pre-defined threshold.

## 6.1 ASYMPTOTIC BEHAVIOR OF COCOA

COCOA defines the following augmentation regions for the control and the treatment groups denoted as $\mathcal{R}_n^0$ and $\mathcal{R}_n^1$ respectively: for $t \in \{0, 1\}$, we have that,

$$\mathcal{R}_n^{1-t} = \{x_j | j \in [n], t_j = 1 - t, \; \exists i_1 < \ldots < i_k \in [n], t_{i_k} = t, \rho(x_{i_k}, x) \leq \epsilon\}$$

where $k$ denotes the number of neighbors. The asymptotic behavior of COCOA is illustrated in the following result.

**Theorem 6.1** (Convergence to RCT). *Let $p_{AF}^1$ and $p_{AF}^0$ be the distributions of the treatment and control groups, respectively, after data augmentation. The following upper bound holds:*

$$V(p_{AF}^1, p_{AF}^0) \leq \frac{1 - \alpha_{n_0}}{1 + z^{-1}\alpha_{n_1}} + \frac{z\alpha_{n_0}(1 - \alpha_{n_1})}{1 + \alpha_{n_0}z} + \frac{|1 - \alpha_{n_0}\alpha_{n_1}|}{(1 + z^{-1}\alpha_{n_1})(1 + \alpha_{n_0}z)}, \tag{8}$$

*Moreover, as $n_1$ and $n_0$ converge to infinity, we have that $1 - \alpha_{n_j}$ converge to 0 with order*

$$1 - \alpha_{n_j} = \mathcal{O}(n_j^k \gamma^{n_j}).$$

This implies that with enough samples, the probability of not encountering data points in close proximity to any given point $x$ becomes very small as the exponential decay $\gamma^{n_j}$ for $\gamma < 1$ dominates. Hence, positivity ensures that within the big data regime, we will encounter densely populated regions, enabling us to approximate counterfactual distributions locally. ***This guarantees that the second term in Theorem( 4.1) converges to zero, thus eliminating the statistical disparity across treatment groups***.

### 6.2 Finite-Sample Guarantee.

Next, we establish the finite-sample guarantees for the GP local regressor. By Mercer's decomposition (Seeger, 2004), a GP is a distribution on a function class $\mathcal{F} \subset \{f : \mathcal{X} \to \mathbb{R}\}$, specified by the GP's kernel $K : \mathcal{X} \times \mathcal{X} \to \mathbb{R}_0^+$.

**Assumption 6.2.** The potential outcome functions belong to this function space $\mathcal{F}$, i.e.,

$$\{f(X, T = t) : \mathcal{X} \to \mathbb{R} \mid t \in \{0, 1\}\} \subset \mathcal{F}.$$

This assumption is reasonable because, with an RBF kernel, $\mathcal{F}$ includes all continuous functions.

**Definition 6.3** (Lipschitz Constant for GP Kernel). Assume that $K : \mathcal{X} \times \mathcal{X} \to \mathbb{R}^+$ is the kernel of a Gaussian Process (GP). Its Lipschitz constant $L_K$ is defined as:

$$L_K(\mathcal{X}) = \sup_{x, x' \in \mathcal{X}} ||\nabla_x K(x, x')||_2. \tag{9}$$

*Remark* 6.4. For well-known kernels, such as RBF, $L_K$ is known and finite if $\mathcal{X}$ is a bounded space. Moreover, $L_K(\mathcal{X})$ is an increasing function of the input space $\mathcal{X}$, i.e., if $\mathcal{X} \subset \mathcal{X}'$, $L_K(\mathcal{X}) \leq L_K(\mathcal{X}')$.

**Data Generation Process.** In this part, we assume that the data generation process is as follows, $Y = f(X, T) + \eta$, where $\eta \sim \mathcal{N}(0, \sigma^2)$ and it is independent of $(X, T)$. We also assume that $\mathcal{X} \subset \mathbb{R}^d$ and the potential outcomes function $f$ are bounded, and $f$ is $L_f$-Lipschitz continuous. Assume there is a dataset $\{x_i, y_i\}_{i=1}^{\bar{n}_t}$ available with $\bar{n}_t$ samples for the imputation of potential outcomes under treatment $t$.

**Imputation Function.** Let $\sigma_{\bar{n}_t}(x) = K(x, x) - K(x, \mathbf{x}_{\bar{n}_t})(K(\mathbf{x}_{\bar{n}_t}, \mathbf{x}_{\bar{n}_t}) + \sigma^2 \cdot I_{\bar{n}_t})^{-1} K(\mathbf{x}_{\bar{n}_t}, x)$ be the posterior standard deviation of GP at $x$ where

$$K(x, \mathbf{x}_{\bar{n}_t}) \in \mathbb{R}^{1 \times \bar{n}_t} = [K(x, x_1), \ldots, K(x, x_{\bar{n}_t})],$$
$$K(\mathbf{x}_{\bar{n}_t}, x) \in \mathbb{R}^{\bar{n}_t \times 1} = [K(x, x_1), \ldots, K(x, x_{\bar{n}_t})]^\top,$$
$$K(\mathbf{x}_{\bar{n}_t}, \mathbf{x}_{\bar{n}_t}) \in \mathbb{R}^{\bar{n}_t \times \bar{n}_t}, K(\mathbf{x}_{\bar{n}_t}, \mathbf{x}_{\bar{n}_t})_{ij} = K(x_i, x_j).$$

Let $\tilde{f}_{\bar{n}_t}(x, t)$ denote the GP-based imputation function given the dataset $\{x_i, y_i\}_{i=1}^{\bar{n}_t} \subset D^t$, i.e., $\tilde{f}_{\bar{n}_t}(x, t) = K(x, \mathbf{x}_{\bar{n}_t})(K(\mathbf{x}_{\bar{n}_t}, \mathbf{x}_{\bar{n}_t}) + \sigma^2 \cdot I_{\bar{n}_t})^{-1} \mathbf{y}_{\bar{n}_t}$ where $\mathbf{y}_{\bar{n}_t} = [y_1, \ldots, y_{\bar{n}_t}]^\top$. Note $\tilde{f}_{\bar{n}_t}$ is a random function, varying with the observed dataset. The following result addresses its error.

**Theorem 6.5.** *For $t \in \{0, 1\}$, let $L_K^t = L_K(\mathcal{R}_n^{1-t})$ denote the Lipschitz constant of the kernel $K$ in region $\mathcal{R}_n^{1-t}$ and let $U_K^t = \sup_{x, x' \in \mathcal{R}_n^{1-t}} K(x, x')$ denote the "width" of region $\mathcal{R}_n^{1-t}$. Then with probability at least $1 - \delta$ where $\delta \in (0, 1)$,*

$$\sup_{t \in \{0,1\}} \sup_{x \in \mathcal{R}_n^{1-t}} |f(x, t) - \tilde{f}_{\bar{n}_t}(x, t)| \leq \sqrt{d} \tilde{\mathcal{O}} \left( \sqrt{\frac{C_K^0 \vee C_K^1}{\bar{n}_0 \wedge \bar{n}_1}} + \sqrt{\sup_{x \in \mathcal{R}_n^1} \sigma_{\bar{n}_0}(x) \vee \sup_{x \in \mathcal{R}_n^0} \sigma_{\bar{n}_1}(x)} \right)$$
$$+ \mathcal{O}(1/(\bar{n}_0 \wedge \bar{n}_1)), \tag{10}$$

*where*

$$C_K^t = 4L_K^t + 2U_K^t/\sigma^2$$

*is only related to the kernel $K$ and unrelated to the number of sample $\bar{n}_t$.*

*Remark* 6.6. Theorem( 6.5) is a sufficient condition for controlling term (III) in Theorem (4.1) due to the fact that

$$\mathbb{E}_{X,T\sim q}\big[\|f(X,T) - \tilde{f}_n(X,T)\|\big] \leq \sup_{t\in\{0,1\}} \sup_{X\in\mathcal{R}_n^{1-t}} |f(X,t) - \tilde{f}_n(X,t)|.$$

*Remark* 6.7. As proved in Theorem 6.1, for any number of required neighbors $\bar{n}_t$, the probability of a fixed $x$ not having more than $\bar{n}_t$ neighbors decreases approximately exponentially to 0. As the right-hand side Equation 10 converges to 0 as $n \to +\infty$, this demonstrates that asymptotically ***COCOA leads to unbiased learning of CATE***.

*Remark* 6.8. COCOA carefully selects the individuals for counterfactual outcome imputation so that:

- By only selecting individuals with a sufficient amount of close neighbors, $\mathcal{R}_n^{1-t}$ is reduced. $\sigma_{\bar{n}_t}(x)$ is also decreased as the posterior of GP has less variance with more close neighbors. Hence, $\sup_{x\in\mathcal{R}_n^{1-t}} \sigma_{\bar{n}_t}(x)$ is significantly reduced, leading to reduced error.

- Smaller $\mathcal{R}_n^{1-t}$ decrease both $L_K^t$ and $U_K^t$, further decreasing the error.

*Remark* 6.9. The effect of the complexity of the true causal function $f$ is captured both in $C_K^t$ and $\sigma_{\bar{n}_t}(x)$: a simpler $f$ implies smoother kernel thus smaller $C_K^t$ and faster decrease of $\sigma_{\bar{n}_t}(x)$.

## 7 EXPERIMENTAL STUDIES

While the theoretical results in Section 6 provide large-sample guarantees, here we empirically demonstrate that COCOA works for practical scenarios where the number of samples is only moderate. In particular, we observe that COCOA consistently improves the CATE estimation performance across state-of-the-art CATE models. More importantly, we observe that ***COCOA prevents CATE models from overfitting to the factual data*** during training. We believe this property is particularly important in the setting of CATE estimation because the true performance of models cannot be validated in practice, making robustness to overfitting an especially desirable property.

**Evaluation Setup.** We test our proposed methods on various benchmark datasets: the IHDP dataset (Ramey et al., 1992; Hill, 2011), the News dataset (Johansson et al., 2016; Newman et al., 2008), and the Twins dataset (Louizos et al., 2017). Additionally, we apply our methods to two synthetic datasets: one with linear functions for potential outcomes and the other with non-linear functions, we include these results in Appendix D.1. A detailed description of these datasets is provided in Appendix B. To estimate the variance of our method, we randomly divide each of these datasets into a train (70%) dataset and a test (30%) dataset with varying seeds. Moreover, we demonstrate the efficacy of our methods across a variety of CATE estimation models.

Table 1: $\sqrt{\varepsilon_{\text{PEHE}}}$ across models, with COCOA augmentation (w/ aug.) and without augmentation (w/o aug.) on Twins, News, and IHDP datasets. Lower $\sqrt{\varepsilon_{\text{PEHE}}}$ corresponds to better performance.

| Model | Twins w/o aug. | Twins w/ aug. | News w/o aug. | News w/ aug. | IHDP w/o aug. | IHDP w/ aug. |
|---|---|---|---|---|---|---|
| TARNet | $0.59_{\pm.29}$ | $0.57_{\pm.32}$ | $5.34_{\pm.34}$ | $5.31_{\pm.17}$ | $0.92_{\pm.01}$ | $0.87_{\pm.01}$ |
| CFR-Wass | $0.50_{\pm.13}$ | $0.14_{\pm.10}$ | $3.51_{\pm.08}$ | $3.47_{\pm.09}$ | $0.85_{\pm.01}$ | $0.83_{\pm.01}$ |
| CFR-MMD | $0.19_{\pm.09}$ | $0.18_{\pm.12}$ | $5.05_{\pm.12}$ | $4.92_{\pm.10}$ | $0.87_{\pm.01}$ | $0.85_{\pm.01}$ |
| T-Learner | $0.11_{\pm.03}$ | $0.10_{\pm.03}$ | $4.79_{\pm.17}$ | $4.73_{\pm.18}$ | $2.03_{\pm.08}$ | $1.69_{\pm.03}$ |
| S-Learner | $0.90_{\pm.02}$ | $0.81_{\pm.06}$ | $3.83_{\pm.06}$ | $3.80_{\pm.06}$ | $1.85_{\pm.12}$ | $0.86_{\pm.01}$ |
| BART | $0.57_{\pm.08}$ | $0.56_{\pm.08}$ | $3.61_{\pm.02}$ | $3.55_{\pm.00}$ | $0.67_{\pm.00}$ | $0.67_{\pm.00}$ |
| CF | $0.57_{\pm.08}$ | $0.51_{\pm.11}$ | $3.58_{\pm.01}$ | $3.56_{\pm.01}$ | $0.72_{\pm.01}$ | $0.63_{\pm.01}$ |

**Performance Improvements.** Table 1 summarizes the experimental results verifying COCOA's effect on ***consistently improving*** the performance of various CATE estimation models. We observe significant improvements for certain models over specific benchmarks (e.g., Twins with CFR-Wass,

Table 2: $\sqrt{\varepsilon_{\text{PEHE}}}$ across different similarity measures: Contrastive Learning (CL), propensity scores (PS), and Euclidean distance (ED), using CFR-Wass across IHDP, News, and Twins datasets.

|       | ED | PS | CL |
|-------|----|----|----|
| IHDP  | $3.32_{\pm 1.13}$ | $3.94_{\pm 0.21}$ | $\mathbf{0.83}_{\pm 0.01}$ |
| News  | $4.98_{\pm 0.10}$ | $4.82_{\pm 0.11}$ | $\mathbf{3.47}_{\pm 0.09}$ |
| Twins | $0.23_{\pm 0.10}$ | $0.48_{\pm 0.09}$ | $\mathbf{0.14}_{\pm 0.10}$ |

IHDP with CD), lead to new state-of-the-art performance. Moreover, even in cases where the improvement is marginal, we note substantial enhancements in models' robustness to overfitting the factual distribution, as described in the following paragraph.

**Robustness Improvements.** In the context of CATE estimation, it is essential to notice the absence of a validation dataset due to the unavailability of the counterfactual outcomes. This poses a challenge in preventing the models from overfitting to the factual distribution. Our proposed data augmentation technique effectively addresses this challenge, as illustrated in Figure 3, resulting in a significant enhancement of the overall effectiveness of various CATE estimation models. Notably, counterfactual balancing frameworks (Johansson et al., 2016; Shalit et al., 2017) significantly benefit from COCOA. This improvement can be attributed to the fact that data augmentation in dense regions helps narrow the discrepancy between the distributions of the control and the treatment groups. By reducing this disparity, our approach enables better generalization and minimizes the balancing distance, leading to more stable outcomes. We include more results in Appendix D.7.

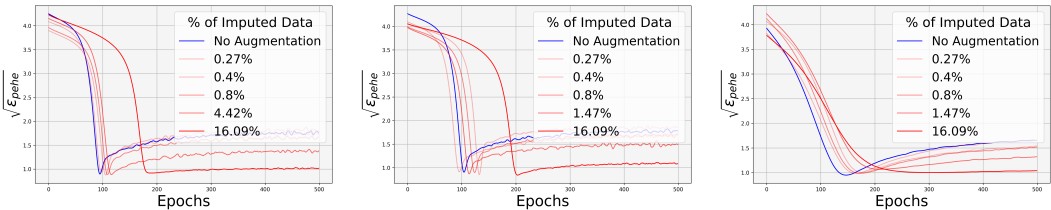

Figure 3: Effects of COCOA on preventing overfitting. From left to right: IHDP with TARNet, CFR-Wass, and T-learner. X-axis has the training epochs; Y-axis shows the performance measure (not accessible in practice). ***The performance of the models trained without data augmentation decreases as the epoch number increases beyond the optimal stopping epoch*** (blue curves), overfitting to the factual distribution. In contrast, ***the error of the models trained with the augmented dataset barely increase*** (red curves), demonstrating the effect of COCOA on preventing overfitting.

**Ablation Studies.** We conducted ablation studies to assess the impact of the embedding ball size ($R$) and the number of neighbors ($k$) on the performance of CATE estimation models trained on the IHDP dataset. Detailed results are in Appendix D.6. These experiments illustrate the trade-off between the quality of imputation and the discrepancy of the treatment groups. ***COCOA is robust to the choice of these hyperparameters***, with a wide range of values leading to performance improvements. Table 2 compares our contrastive learning method to propensity scores and Euclidean distance as similarity measures. Appendix D.4 includes ATE estimation results, and Appendix D.5 covers ablations on GP and local linear regression kernels.

## 8 CONCLUSION

In this paper, we present a model-agnostic data augmentation method for CATE estimation. We propose a generalization bound motivating our approach. We utilize contrastive learning and Gaussian Processes to reliably impute some missing counterfactuals. We provide both asymptotic and finite sample guarantees to support the proposed method. Notably, we enhance the performance and robustness of various CATE estimation models across various datasets.

**Ethics Statement.** This work focuses on improving the design of machine learning models for estimating treatment effects. We do not foresee any immediate ethical concerns.

**Reproducibility Statement.** We have provided detailed information on how the datasets are processed and how the models are trained, including hyperparameters values. Additionally, we have included the implementation of our algorithms in Python in the supplementary material.

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

## A    APPENDIX

## B    DATASET DESCRIPTIONS

**IHDP**    The IHDP dataset is a semi-synthetic dataset that was introduced based on real covariates available from the Infant Health and Development Program (IHDP) to study the effect of development programs on children. The features (covariates) in this dataset come from a Randomized Control Trial. The potential outcomes were simulated following Setting B in Hill (2011). The IHDP dataset consists of 747 individuals (139 in the treatment group and 608 in the control group), each with 25 features. The potential outcomes are generated as follows:

$$Y_0 \sim \mathcal{N}(\exp(\beta^T (X + W)), 1)$$

and

$$Y_1 \sim \mathcal{N}(\beta^T (X + W) - \omega, 1)$$

where $W$ has the same dimension as $X$ with all entries equal $0.5$ and $\omega = 4$. The regression coefficient $\beta$ is a vector of length 25 where each element is randomly sampled from a categorical distribution with the support $(0, 0.1, 0.2, 0.3, 0.4)$ and the respective probability masses $\mu = (0.6, 0.1, 0.1, 0.1, 0.1)$.

**News**    The News Dataset is a semi-synthetic dataset designed to assess the causal effects of various news topics on reader responses. It was first introduced in Johansson et al. (2016). The documents were sampled from news items from the NY Times corpus (downloaded from UCI Newman et al. (2008)). The covariates available for CATE estimation are the raw word counts for the 100 most probable words in each topic. The treatment $t \in \{0, 1\}$ denotes the viewing device. $t = 0$ means *with computer* and $t = 1$ means *with mobile*. A topic model is trained on a comprehensive collection of documents to generate $z(x) \in \mathbb{R}^k$ that represents the topic distribution of a given news item $x$ (Johansson et al., 2016).

Let the treatment effects be represented by $z_{c_1}$ (for $t = 1$) and $z_{c_0}$ (for $t = 0$) $z_{c_1}$ is defined as the topic distribution of a randomly selected document while $z_{c_0}$ is the average topic representation across all documents. The reader's opinion of news item $x$ on device $t$ is influenced by the similarity between $z(x)$ and $z_{c_t}$, expressed as:

$$y(x, t) = C \cdot \left( z(x)^T z_{c_0} + t \cdot z(x)^T z_{c_1} \right) + \epsilon$$

where $C = 50$ is a scaling factor and $\epsilon \sim \mathcal{N}(0, 1)$. The assignment of a news item $x$ to a device $t \in \{0, 1\}$ is biased towards the preferred device for that item, modeled using the softmax function:

$$p(t = 1 | x) = \frac{e^{\kappa \cdot z(x)^T z_{c_1}}}{e^{\kappa \cdot z(x)^T z_{c_0}} + e^{\kappa \cdot z(x)^T z_{c_1}}}$$

Here, $\kappa$ determines the strength of the bias and it is assigned to be 10.

**Twins** The Twins dataset Louizos et al. (2017) is based on the collected birthday data of twins born in the United States from 1989 to 1991. It is assumed that twins share significant parts of their features. Consider the scenario where one of the twins was born heavier than the other as the treatment assignment. The outcome is whether the baby died in infancy (i.e., the outcome is mortality). Here, the twins are divided into two groups: the treatment and the control groups. The treatment group consists of heavier babies from the twins. On the other hand, the control group consists of lighter babies from the twins. The potential outcomes, $Y_0$ and $Y_1$, are generated through:

$$Y_0 \sim \mathcal{N}(\exp(\beta^T X), 0.2)$$

and

$$Y_1 \sim \mathcal{N}(\alpha^T X, 0.2)$$

Where $\beta$ and $\alpha$ are sampled from a high dimensional standard normal distribution.

**Linear dataset** We synthetically generate a dataset with $N = 1500$ samples and $d = 10$ features. The feature vectors $X = (x_1, x_2, \ldots, x_d)^T \in \mathbb{R}^d$ are drawn from a standard normal distribution. The treatment assignment $t \in \{0, 1\}$ is biased, with the probability of treatment being

$$p(t = 1|x) = \frac{1}{1 + \exp(-(x_1 + x_2))}$$

We generate potential outcomes using two linear functions with coefficients $\beta_0 = (0.5, , \ldots, 0.5) \in \mathbb{R}^d$ and $\beta_1 = (0.3, \ldots, 0.3) \in \mathbb{R}^d$ as follows:

$$Y_0 = \beta_0 X + \mathcal{N}(0, 0.01)$$
$$Y_1 = \beta_1 X + \mathcal{N}(0, 0.01)$$

**Non-Linear dataset** We construct a synthetic dataset consisting of $N = 1500$ instances with $d = 10$ features. The feature vectors, denoted by $X = (x_1, x_2, \ldots, x_d)^T \in \mathbb{R}^d$, are sampled from a standard normal distribution. The treatment assignment $t \in \{0, 1\}$ is biased, with the probability of treatment being

$$p(t = 1|x) = \frac{1}{1 + \exp(-(x_1 + x_2))}$$

We generate potential outcomes using two linear functions with coefficients $\beta_0 = (0.5, , \ldots, 0.5) \in \mathbb{R}^d$ and $\beta_1 = (0.3, \ldots, 0.3) \in \mathbb{R}^d$ as follows:

$$Y_0 = \exp(\beta_0 X) + \mathcal{N}(0, 0.01)$$
$$Y_1 = \exp((\beta_1 X) + \mathcal{N}(0, 0.01)$$

## C PROOFS OF THE THEORETICAL RESULTS

In this section, we include the proofs for the theoretical results presented in the main text.

### C.1 DISTRIBUTION OF THE AUGMENTED DATASET

The marginal distribution of $(X, T)$ in the augmented dataset can be defined as follows:

$$p_{\text{AF}}(x, t) = \frac{1}{1 + \beta} p_{\text{F}}(x, t) + \frac{\beta}{1 + \beta} q(x, 1 - t),$$

where $\frac{\beta}{1 + \beta} \in [0, \frac{1}{2}]$ represents the ratio of the number of the select individuals for augmentation to the total number of samples in the augmented dataset, and $q = \frac{p_{\text{F}}(x, 1-t)}{\alpha} \mathbb{1}_{\mathcal{R}_n}$, with $\alpha$ as the normalizing constant, i.e., $\alpha = \int p_{\text{F}}(x, 1 - t) \mathbb{1}_{\mathcal{R}_n}(x, 1 - t) dx dt$. In other words, $q$ is the factual distribution of the alternative treatment group *with its probability mass normalized to the augmentation region $\mathcal{R}_n$*.

Hence, $p_{\text{AF}}(y|x, t)$ can be defined as follows: it is equal to $p_{\text{F}}(y|x, t)$ when $(x, t)$ is sampled from the factual distribution; for samples drawn from $q(x, 1 - t)$, $p_{\text{AF}}(y|x, t)$ is defined as a point mass function $\delta(y = \tilde{f}_n(x, t))$.

## C.2   PROOF OF THEOREM( 4.1)

**Theorem 4.1.** Let $h$ be a hypothesis, its $\varepsilon_{\text{PEHE}}$ is upper bounded as follows:

$$\varepsilon_{\text{PEHE}}(h) \leq 4 \cdot \left( \underbrace{\mathcal{L}_{p_{\text{AF}}}(h)}_{\text{(I)}} + 2\underbrace{V\left(p_{\text{RCT}}(X,T), p_{\text{AF}}(X,T)\right)}_{\text{(II)}} + \underbrace{\frac{\beta}{1+\beta} \cdot b_{\mathcal{A}}(n)}_{\text{(III)}} \right)$$

where $V(g_1, g_2) = \frac{1}{2} \int_{\mathcal{S}} |g_1(s) - g_2(s)| ds$ is the total variation distance, and

$$b_{\mathcal{A}}(n) = \mathbb{E}_{X,T \sim q}\left[\|f(X,T) - \tilde{f}_n(X,T)\|^2\right]$$

To prove the generalization bound, we first define a notion of consistency for data augmentation. And, we demonstrate a lemma proving that the proposed consistency is equivalent to emulating RCTs.

**Definition C.1** (Consistency of Factual Distribution). A factual distribution $p_F$ is consistent if for every hypothesis $h : \mathcal{X} \times \{0,1\} \to \mathcal{Y}, \mathcal{L}_{\text{F}}(h) = \mathcal{L}_{\text{CF}}(h)$.

**Definition C.2** (Consistency of Data Augmentation). A data augmentation method is said to be consistent if the augmented data follows a factual distribution that is consistent.

**Lemma C.3** (Consistency is Equivalent Randomized Controlled Trials). *Suppose we have a factual distribution $p_F$ and its corresponding counterfactual distribution $p_{CF}$ such that for every hypothesis $h : \mathcal{X} \times \{0,1\} \to \mathcal{Y}, \mathcal{L}_F(h) = \mathcal{L}_{CF}(h)$. This implies that the data must originate from a randomized controlled trial, i.e., $p_F(X|T = 1) = p_F(X|T = 0)$.*

*Proof of Lemma C.3.*
Suppose that for every hypothesis $h : \mathcal{X} \times \{0,1\} \to \mathcal{Y}, \mathcal{L}_{\text{F}}(h) = \mathcal{L}_{\text{CF}}(h)$.
By definition,

$$\mathcal{L}_{\text{F}}(h) = \int (y - h(x,t))^2 p_{\text{F}}(x,t,y) \, dx \, dt \, dy$$

and

$$\mathcal{L}_{\text{CF}}(h) = \int (y - h(x,t))^2 p_{\text{CF}}(x,t,y) \, dx \, dt \, dy$$

We can write this as

$$\mathbb{E}_{p_{\text{F}}}\left[\left(Y - h(X,T)^2\right)\right] = \mathbb{E}_{p_{\text{CF}}}\left[\left(Y - h(X,T)^2\right)\right]$$

Since this holds for every function $h$, consider two Borel sets A and B in $\mathcal{X} \times \mathcal{T} \times \mathcal{Y}$, and we let $h_1(X,T) = \mathbb{E}[Y|X,T] - \mathbb{1}_A$ and $h_2(X,T) = \mathbb{E}[Y|X,T] - \mathbb{1}_B$. Hence we have that,

$$\mathbb{E}_{p_{\text{F}}}\left[(Y - h_1(X,T))^2\right] = \mathbb{E}_{p_{\text{F}}}\left[(Y - \mathbb{E}[Y|X,T] + \mathbb{1}_A)^2\right]$$

$$= \mathbb{E}_{p_{\text{F}}}\left[(Y - \mathbb{E}[Y|X,T])^2\right] + \mathbb{E}_{p_{\text{F}}}[\mathbb{1}_A] + 2\mathbb{E}_{p_{\text{F}}}[\mathbb{1}_A(Y - \mathbb{E}[Y|X,T])]$$

And we have that, $\mathbb{E}_{p_{\text{F}}}[\mathbb{1}_A(Y - \mathbb{E}[Y|X,T])] = 0$ since by definition of the conditional expectation we have that $\mathbb{E}[Y\mathbb{1}_A] = \mathbb{E}[\mathbb{E}[Y|X,T]\mathbb{1}_A]$. We denote by $MSE(p_{\text{F}}) = \mathbb{E}_{p_{\text{F}}}\left[(Y - \mathbb{E}[Y|X,T])^2\right]$. Therefore we have that

$$\mathbb{E}_{p_{\text{F}}}\left[(Y - h_1(X,T))^2\right] = MSE(p_{\text{F}}) + \mathbb{E}_{p_{\text{F}}}[\mathbb{1}_A]$$

Using the same argument for $p_{\text{CF}}$ we have the following result:

$$\mathbb{E}_{p_{\text{CF}}}\left[(Y - h_1(X,T))^2\right] = MSE(p_{\text{CF}}) + \mathbb{E}_{p_{\text{CF}}}[\mathbb{1}_A]$$

Similarly, we have the following for $h_2$:

$$\mathbb{E}_{p_{\text{F}}}\left[(Y - h_2(X,T))^2\right] = MSE(p_{\text{F}}) + \mathbb{E}_{p_{\text{F}}}[\mathbb{1}_B]$$

$$\mathbb{E}_{p_{\text{CF}}}\left[(Y - h_2(X,T))^2\right] = MSE(p_{\text{CF}}) + \mathbb{E}_{p_{\text{CF}}}[\mathbb{1}_B]$$

Therefore we have

$$MSE(p_{\text{F}}) - MSE(p_{\text{CF}}) = \mathbb{E}_{p_{\text{F}}}\left[\mathbb{1}_A\right] - \mathbb{E}_{p_{\text{CF}}}\left[\mathbb{1}_A\right]$$

and

$$MSE(p_{\text{F}}) - MSE(p_{\text{CF}}) = \mathbb{E}_{p_{\text{F}}}\left[\mathbb{1}_B\right] - \mathbb{E}_{p_{\text{CF}}}\left[\mathbb{1}_B\right]$$

Therefore

$$\mathbb{E}_{p_{\text{F}}}\left[\mathbb{1}_A\right] - \mathbb{E}_{p_{\text{CF}}}\left[\mathbb{1}_A\right] = \mathbb{E}_{p_{\text{F}}}\left[\mathbb{1}_B\right] - \mathbb{E}_{p_{\text{CF}}}\left[\mathbb{1}_B\right]$$

Hence it follows,

$$\mathbb{E}_{p_{\text{F}}}\left[\mathbb{1}_{A\cap B}\right] = \mathbb{E}_{p_{\text{CF}}}\left[\mathbb{1}_{A\cap B}\right]$$

And as this holds for every Borel measurable set $A$ and $B$, therefore we have that $p_{\text{F}} = p_{\text{CF}}$.

Denote by $u = p_{\text{F}}(T = 1)$ we have $p_{\text{F}}(X) = u p_{\text{F}}(X|T = 1) + (1 - u)p_{\text{F}}(X|T = 0)$. Similarly we have that $p_{\text{CF}}(X) = (1 - u)p_{\text{CF}}(X|T = 1) + u p_{\text{CF}}(X|T = 0)$. Therefore, since $p_{\text{F}} = p_{\text{CF}}$,

$$
\begin{aligned}
u p_{\text{F}}(X|T = 1) + (1 - u)p_{\text{F}}(X|T = 0) &= (1 - u)p_{\text{CF}}(X|T = 1) + u p_{\text{CF}}(X|T = 0) \\
&= (1 - u)p_{\text{F}}(X|T = 1) + u p_{\text{F}}(X|T = 0)
\end{aligned}
$$

Hence

$$(2u - 1)\, p_{\text{F}}(X|T = 1) = (2u - 1)\, p_{\text{F}}(X|T = 0)$$

Therefore we conclude the result that,

$$p_{\text{F}}(X|T = 1) = p_{\text{F}}(X|T = 0).$$

This concludes the proof. $\qquad\square$

For completeness, we also include this result.

**Lemma C.4** (Consistency of Randomized Controlled Trials). *The factual distribution of any randomized controlled trial =verifying $p_F(T = 1) = p_F(T = 0)$ is consistent, i.e., if $p_F(X|T = 1) = p_F(X|T = 0)$ and $p_F(T = 1) = p_F(T = 0)$, then for all $h : \mathcal{X} \times \{0, 1\} \to \mathcal{Y}$,*

$$\mathcal{L}_F(h) = \mathcal{L}_{CF}(h)$$

*Proof.* Let $u = p_F(T = 1) = \frac{1}{2}$, $p_F(T = 1) = p_{CF}(T = 0)$

$$
\begin{aligned}
\mathcal{L}_{\text{F}}(h) \\
&= \int (y - h(x, t))^2 p_{\text{F}}(x, t, y)\, dx,\, dt\, dy \\
&= u \int (y - h(x, 1))^2 p_{\text{F}}(x, y|T = 1)\, dx\, dy + (1 - u)\int (y - h(x, 0))^2 p_{\text{F}}(x, y|T = 0)\, dx\, dy \\
&= u \int (y - h(x, 1))^2 p_{\text{F}}(x, y|T = 0)\, dx\, dy + (1 - u)\int (y - h(x, 0))^2 p_{\text{F}}(x, y|T = 1)\, dx\, dy \\
&= u \int (y - h(x, 1))^2 p_{\text{CF}}(x, y|T = 1)\, dx\, dy + (1 - u)\int (y - h(x, 0))^2 p_{\text{CF}}(x, y|T = 0)\, dx\, dy \\
&= \int (y - h(x, t))^2 p_{\text{CF}}(x, t, y)\, dx\, dy \\
&= \mathcal{L}_{\text{CF}}(h)
\end{aligned}
$$

$\qquad\square$

To prove Theorem (4.1) we also include a new definition for an "ideal" factual distribution. Subsequently, we will prove its consistency. The ideal factual distribution is defined as follows:

$$p_{\text{IF}} = \frac{1}{2}p_{\text{F}} + \frac{1}{2}p_{\text{CF}}. \tag{11}$$

In other words, to sample a dataset from $p_{\text{IF}}$, we sample from the factual distribution $p_{\text{F}}$ half of the time and from the counterfactual distribution $p_{\text{CF}}$ in the other half of the times. Let $p_{\text{ICF}}$ denote the counterfactual distribution corresponding to $p_{\text{IF}}$. We next show that $p_{\text{IF}}$ is consistent (thus called ideal distribution).

**Lemma C.5** (Consistency of $p_{IF}$.). *The error of the ideal factual distribution equals the error of its corresponding counterfactual distribution, i.e., for every hypothesis $h : \mathcal{X} \times \{0,1\} \to \mathcal{Y}$, we have that $\mathcal{L}_{IF}(h) = \mathcal{L}_{ICF}(h)$.*

*Proof.* We observe that $p_{ICF} = \frac{1}{2}p_{CF} + \frac{1}{2}p_{F}$. Therefore, $p_{ICF} = p_{IF}$ and the result follows. $\square$

Intuitively, this result is saying that the ideal counterfactual augmentation gives us a factual distribution that perfectly balances the factual and counterfactual worlds. It follows from Lemma C.3 that achieving this property guarantees that the dataset is identically distributed to the one generated from a Randomized Controlled Trial. However, it is impossible to sample from $p_{CF}$.

Also, we cite this Theorem that we will use in our proof:

**Theorem C.6** (Theorem 1 in Ben-David et al. (2010)). *Let $f$ be the true function for a learning task such that $f(x) = \mathbb{E}[Y|X = x]$ where $X$ has a density $p$ and let another true function $g(x) = \mathbb{E}[Y|X = x]$ modeling another learning task, where $X$ has a density $q$. Let $h$ by a hypothesis function estimating the true function $f$, therefore we have*

$$\mathbb{E}_{X \sim q(x)}[\|g(X) - h(X)\|^2] \leq \mathbb{E}_{X \sim p(x)}[\|f(X) - h(X)\|^2] + 2V(p(x), p(x))$$
$$+ \mathbb{E}_{X \sim p(x)}[\|f(X) - g(X)\|^2]$$

We can now prove Theorem( 4.1).

*Proof.* We have $f : \mathcal{X} \times \{0,1\} \to \mathcal{Y}$ to be the function underlying the true causal relationship between $(X,T)$ and $Y$. It follows from Theorem C.6 that:

$$\mathcal{L}_{IF}(h) \leq \mathcal{L}_{AF}(h) + 2V(p_{IF}, p_{AF}) + \mathbb{E}_{x,t \sim p_{AF}}[\|f(x,t) - \tilde{f}(x,t)\|^2]$$

where $\mathcal{L}_{IF}$ is the factual loss with respect to the ideal density and $\mathcal{L}_{AF}$ is the factual loss with respect to the density of the augmented data.

By decomposition of the $\varepsilon_{PEHE}$ we have that,

$$\varepsilon_{PEHE}(h) = \int_{\mathcal{X}} (h(x,1) - h(x,0) - f(x,1) + f(x,0))^2 \, p_{IF}(x) dx$$

$$= \int_{\mathcal{X}} (h(x,1) - h(x,0) - f(x,1) + f(x,0))^2 \, p_{IF}(x|T=1) p(T=1) dx dt$$

$$+ \int_{\mathcal{X}} (h(x,1) - h(x,0) - f(x,1) + f(x,0))^2 \, p_{IF}(x|T=0) p(T=0) dx dt$$

$$\leq 2 \cdot \mathcal{L}_{IF}(h) + 2 \cdot \mathcal{L}_{ICF}(h)$$

Therefore, it follows from Lemma C.5 that,

$$\varepsilon_{PEHE}(h) \leq 4 \cdot \left( \mathcal{L}_{AF}(h) + 2V(p_{RCT}(x,t), p_{AF}(x,t)) + \mathbb{E}_{x,t \sim p_{AF}}[\|f(x,t) - \tilde{f}_n(x,t)\|^2] \right)$$

And since we have that,

$$\mathbb{E}_{x,t \sim p_{AF}}[\|f(x,t) - \tilde{f}_n(x,t)\|^2]) =$$
$$(\frac{1}{1+\beta}) \cdot \mathbb{E}_{x,t \sim p_F}[\|f(x,t) - \tilde{f}_n(x,t)\|] + \cdot \frac{\beta}{1+\beta} \mathbb{E}_{x,t \sim q}[\|f(x,t) - \tilde{f}_n(x,t)\|]$$

And by observing that the first term $\mathbb{E}_{x,t \sim p_F}[\|f(x,t) - \tilde{f}_n(x,t)\|^2] = 0$, since the algorithm keeps the samples from the factual distribution to be the same. $\square$

## C.3   PROOF OF THEOREM( 6.1)

*__Theorem 6.1__*. Let $p_{AF}^1$ and $p_{AF}^0$ be the distributions of the treatment and control groups, respectively, after data augmentation. The following upper bound holds:

$$V(p_{AF}^1, p_{AF}^0) \leq \frac{1 - \alpha_{n_0}}{1 + z^{-1}\alpha_{n_1}} + \frac{z\alpha_{n_0}(1 - \alpha_{n_1})}{1 + \alpha_{n_0}z} + \frac{|1 - \alpha_{n_0}\alpha_{n_1}|}{(1 + z^{-1}\alpha_{n_1})(1 + \alpha_{n_0}z)},$$

as $n_1$ and $n_0$ converge to infinity, we have that $\alpha_{n_1}$ and $\alpha_{n_0}$ converge to 1 with $1 - \alpha_{n_j} = \mathcal{O}(n_j^k \gamma^{n_j})$.

To prove Theorem( 6.1) we start by stating the following lemma and proving it.

**Lemma C.7.** *Let $a, b \in [0, 1]$, and let $(p_1, p_2, q_1, q_2)$ be probability distributions, we have that:*

$$V(ap_1 + (1 - a)p_2, bq_1 + (1 - b)q_2) \leq a \cdot V(p_1, q_1) + b \cdot V(p_2, q_2) + |a - b|.$$

*Proof.* Let $a, b \in [0, 1]$, and let $(p_1, p_2, q_1, q_2)$ be probability distributions, we have that:

$$V(ap_1 + (1 - a)p_2, bq_1 + (1 - b)q_2)$$

$$= \frac{1}{2} \int_{\mathbb{R}^d} |ap_1(x) + (1 - a)p_2(x) - bq_1(x) + (1 - b)q_2(x)| \, dx$$

$$\leq \frac{1}{2} \int_{\mathbb{R}^d} |ap_1(x) - bq_1(x)| + |(1 - a)p_2(x) - (1 - b)q_2(x)| \, dx$$

With triangle inequality again, we can bound

$$|ap_1(x) - bq_1(x)| \leq a \, |p_1(x) - q_1(x)| + |a - b| \, |q_1(x)|$$

and,

$$|(1 - a)p_2(x) - (1 - b)q_2(x)| \leq (1 - a) \, |p_2(x) - q_2(x)| + |a - b| \, |q_2(x)|$$

and by integrating we have that,

$$V(ap_1 + (1 - a)p_2, bq_1 + (1 - b)q_2) \leq a \cdot V(p_1, q_1) + b \cdot V(p_2, q_2) + |a - b|.$$

$\square$

*Proof.* We start by proving the rate of convergence. We have that,

$$\mathbb{P}(X^0 \in \mathcal{R}_n^0) = \sum_{i=k}^{n_1} \binom{n_1}{i} (1 - \gamma)^i \gamma^{n_1 - i}$$

$$= 1 - \sum_{i=0}^{k-1} \binom{n_1}{i} (1 - \gamma)^i \gamma^{n_1 - i}$$

$$= 1 - \sum_{i=1}^{k-1} \frac{n_1!}{(n_1 - i)!i!} (1 - \gamma)^i \gamma^{n_1 - i}$$

$$\geq 1 - \frac{n_1!}{(n_1 - k + 1)!} \gamma^{n_1} \sum_{i=0}^{k-1} \frac{1}{i!} \left( \frac{1 - \gamma}{\gamma} \right)^i$$

$$\geq 1 - n_1^k \gamma^{n_1} \sum_{i=0}^{k-1} \frac{1}{i!} \left( \frac{1 - \gamma}{\gamma} \right)^i$$

Similarly, we have,

$$\mathbb{P}(X^1 \in \mathcal{R}_n^1) = \sum_{i=k}^{n_0} \binom{n_0}{i} (1 - \gamma)^i \gamma^{n_0 - i}$$

$$= 1 - \sum_{i=1}^{k-1} \frac{n_0!}{(n_0 - i)!i!} (1 - \gamma)^i \gamma^{n_0 - i}$$

$$\geq 1 - n_0^k \gamma^{n_0} \sum_{i=0}^{k-1} \frac{1}{i!} \left( \frac{1 - \gamma}{\gamma} \right)^i$$

Therefore we have,

$$1 - \alpha_{n_0} = \mathcal{O}(n_0^k \gamma^{n_0}),$$

$$1 - \alpha_{n_1} = \mathcal{O}(n_1^k \gamma^{n_1}),$$

We now state the definition of the probability densities of the control and treatment groups resulting from the augmentation process as,

$$p_{\text{AF}}^1 = \frac{1}{1 + \beta_{n_1}} p^1 + \frac{\beta_{n_1}}{1 + \beta_{n_1}} \frac{p^0 \mathbb{1}_{\mathcal{R}_0}}{\alpha_{n_1}}$$

and,

$$p_{\text{AF}}^0 = \frac{1}{1 + \beta_{n_0}} p^0 + \frac{\beta_{n_0}}{1 + \beta_{n_0}} \frac{p^1 \mathbb{1}_{\mathcal{R}_1}}{\alpha_{n_0}}$$

with,

$$\beta_{n_1} = \alpha_{n_1} \left( \frac{1 - u}{u} \right)$$

and,

$$\beta_{n_0} = \alpha_{n_0} \left( \frac{u}{1 - u} \right)$$

$$V(p_{\text{AF}}^1, p_{\text{AF}}^0) = \frac{1}{2} \int |p_{\text{AF}}^1 - p_{\text{AF}}^0|$$

$$= \frac{1}{2} \int \left| \frac{1}{1 + \beta_{n_1}} p^1 + \frac{\beta_{n_1}}{1 + \beta_{n_1}} \frac{p^0 \mathbb{1}_{\mathcal{R}_0}}{\alpha_{n_1}} - \frac{1}{1 + \beta_{n_0}} p^0 - \frac{\beta_{n_0}}{1 + \beta_{n_0}} \frac{p^1 \mathbb{1}_{\mathcal{R}_1}}{\alpha_{n_0}} \right|$$

$$\leq \frac{1}{2} \int \left| \frac{1}{1 + \beta_{n_1}} p^1 - \frac{\beta_{n_0}}{1 + \beta_{n_0}} \frac{p^1 \mathbb{1}_{\mathcal{R}_1}}{\alpha_{n_0}} \right| + \frac{1}{2} \int \left| \frac{\beta_{n_1}}{1 + \beta_{n_1}} \frac{p^0 \mathbb{1}_{\mathcal{R}_0}}{\alpha_{n_1}} - \frac{1}{1 + \beta_{n_1}} p^0 \right|$$

Hence by applying Lemma C.7 we have that,

$$V(p_{\text{AF}}^1, p_{\text{AF}}^0) \leq \frac{1}{1 + \beta_{n_1}} V(p^1, \frac{p^1 \mathbb{1}_{\mathcal{R}_1}}{\alpha_{n_0}}) + \frac{\beta_{n_0}}{1 + \beta_{n_0}} V(p^0, \frac{p^0 \mathbb{1}_{\mathcal{R}_0}}{\alpha_{n_0}}) + |\frac{1}{1 + \beta_{n_1}} - \frac{\beta_{n_0}}{1 + \beta_{n_0}}|$$

We have that,

$$V(p^1, \frac{p^1 \mathbb{1}_{\mathcal{R}_1}}{\alpha_{n_0}}) = \frac{1}{2} \left( \int_{\mathcal{R}_1} |p^1 - \frac{p^1}{\alpha_{n_0}}| + \int_{\mathcal{R}_1^c} p^1 \right)$$

$$= \frac{1}{2} \left( \int_{\mathcal{R}_1} p^1 |1 - \frac{1}{\alpha_{n_0}}| + (1 - \alpha_{n_0}) \right)$$

$$= \frac{1}{2} \left( \frac{|\alpha_{n_0} - 1|}{\alpha_{n_0}} \int_{\mathcal{R}_1} p^1 + (1 - \alpha_{n_0}) \right)$$

$$= \frac{1}{2} \left( \frac{|\alpha_{n_0} - 1|}{\alpha_{n_0}} \alpha_{n_0} + (1 - \alpha_{n_0}) \right)$$

$$= (1 - \alpha_{n_0})$$

Similarly,

$$V(p^0, \frac{p^0 \mathbb{1}_{\mathcal{R}_0}}{\alpha_{n_1}}) = (1 - \alpha_{n_1})$$

Substituting this into the bound and letting $z = \frac{u}{1 - u}$ we have that,

$$V(p_{\text{AF}}^1, p_{\text{AF}}^0) \leq \frac{1 - \alpha_{n_0}}{1 + \beta_{n_1}} + \frac{\beta_{n_0}(1 - \alpha_{n_1})}{1 + \beta_{n_0}} \alpha_{n_1}) + |\frac{1}{1 + \beta_{n_1}} - \frac{\beta_{n_0}}{1 + \beta_{n_0}}|$$

$$= \frac{1 - \alpha_{n_0}}{1 + z^{-1} \alpha_{n_1}} + \frac{z \alpha_{n_0} (1 - \alpha_{n_1})}{1 + \alpha_{n_0} z} + \frac{|1 - \alpha_{n_1} \alpha_{n_0}|}{(1 + z^{-1} \alpha_{n_1})(1 + \alpha_{n_0} z)}$$

$$\square$$

### C.4 PROOF OF THEOREM( 6.5)

**Theorem 6.5.** For $t \in \{0,1\}$, let $L_K^t = L_K(\mathcal{R}_n^{1-t})$ denote the Lipschitz constant of the kernel $K$ in region $\mathcal{R}_n^{1-t}$ and let $U_K^t = \sup_{x,x' \in \mathcal{R}_n^{1-t}} K(x,x')$ denote the "width" of region $\mathcal{R}_n^{1-t}$. Then with probability at least $1 - \delta$ where $\delta \in (0,1)$,

$$\sup_{t \in \{0,1\}} \sup_{x \in \mathcal{R}_n^{1-t}} |f(x,t) - \tilde{f}_{\bar{n}_t}(x,t)| \leq \sqrt{d} \tilde{\mathcal{O}} \left( \sqrt{\frac{C_K^0 \vee C_K^1}{\bar{n}_0 \wedge \bar{n}_1}} + \sqrt{\sup_{x \in \mathcal{R}_n^1} \sigma_{\bar{n}_0}(x) \vee \sup_{x \in \mathcal{R}_n^0} \sigma_{\bar{n}_1}(x)} \right)$$
$$+ \mathcal{O}(1/(\bar{n}_0 \wedge \bar{n}_1)),$$

where

$$C_K^t = 4L_K^t + 2U_K^t/\sigma^2$$

is only related to the kernel $K$ and unrelated to the number of sample $\bar{n}_t$.

*Proof.* The proof for $t = 0$ and $t = 1$ is symmetric, thus fix $t \in \{0,1\}$. For notational simplicity, we use $z$ in the proof to denote $\bar{n}_t$, and let

$$A = (K(\mathbf{x}_z, \mathbf{x}_z) + \sigma^2 \cdot I_z)^{-1} \in \mathbb{R}^{z \times z}.$$

and

$$U_K^t = \max_{x,x' \in \mathcal{R}_n^t} K(x,x').$$

Consider $\tau > 0$. A set $S$ is a $\tau$-*cover* for $\mathcal{R}_n^{1-t}$ if $\forall x \in \mathcal{R}_n^{1-t}, \exists x' \in S$ such that $||x' - x|| \leq \tau$. Let $\mathcal{C}(\tau, \mathcal{R}_n^{1-t})$ be the covering number of $\mathcal{R}_n^{1-t}$ with radius $\tau$:

$$\mathcal{C}(\tau, \mathcal{R}_n^{1-t}) \doteq \inf\{|S| : S \text{ is } \tau\text{-cover of } \mathcal{R}_n^{1-t}\}.$$

Since $\mathcal{R}_n^{1-t} \subset \mathbb{R}^d$, we have Vaart & Wellner (2023)

$$\mathcal{C}(\tau, \mathcal{R}_n^{1-t}) \leq \left(1 + \frac{r}{\tau}\right)^d,$$

where $r \doteq \max_{x,x' \in \mathcal{R}_n^{1-t}} ||x - x'||$. Consider a minimum $\tau$-cover $\mathcal{C}_\tau$ for $\mathcal{R}_n^{1-t}$ with (by definition of covering number) $\mathcal{C}(\tau, \mathcal{R})$ elements. We have that Srinivas et al. (2012), with probability at least $1 - \mathcal{C}(\tau, \mathcal{R}) \exp(-\xi(\tau)/2)$,

$$\sup_{x \in \mathcal{C}_\tau} |f(x,t) - \tilde{f}_n(x,t)| \leq \sqrt{\xi(\tau)} \sup_{x \in \mathcal{C}_\tau} \sigma_n(x).$$

Choosing $\xi(\tau) = 2 \log(\mathcal{C}(\tau, \mathcal{R})/\delta)$, we have with probability $1 - \delta$,

$$\sup_{x \in \mathcal{C}_\tau} |f(x,t) - \tilde{f}_n(x,t)| \leq \sqrt{\xi(\tau)} \sup_{x \in \mathcal{C}_\tau} \sigma_n(x).$$

Moreover, by definition of $\mathcal{C}_\tau$, $\max_{x \in \mathcal{R}_n^t} \min_{x' \in \mathcal{C}_\tau} ||x - x'|| \leq \tau$. Because $f(x,t)$ is $L_f$-Lipschitz continuous, we have for all $x \in \mathcal{R}_n^{1-t}$

$$\min_{x' \in \mathcal{C}_\tau} |f(x,t) - f(x',t)| \leq \tau L_f.$$

With the fact that Lederer et al. (2019) $\tilde{f}_z(x,t)$ and $\sigma_z(x)$ is Lipschitz continuous with respective Lipschitz constant

$$C_1 = L_K \sqrt{z} ||A\mathbf{y}_n||, \tag{12}$$

$$C_2(\tau) = \sqrt{2\tau L_K (1 + z \cdot ||A||_F \cdot U_K^t)}, \tag{13}$$

we have with probability at least $1 - \delta$ that

$$\sup_{x \in \mathcal{R}_n^{1-t}} |\tilde{f}_z(x,t) - f(x,t)| \leq \sqrt{\xi(\tau)} \sup_{x \in \mathcal{R}_n^{1-t}} \sigma_z(x) + C_2(\tau)\sqrt{\xi(\tau)} + (C_1 + L_f)\tau$$

To continue, we will proceed to upper bound $C_1$:

$$C_1 = L_K \sqrt{z} ||A\mathbf{y}_z|| \leq L_K \sqrt{z} ||A||_F ||\mathbf{y}_z|| \leq L_K \sqrt{z} \frac{||\mathbf{y}_z||}{\sigma^2}$$

due to the fact that $||A||_F \leq 1/\sigma^2$. Assume that $f(x,t) \leq F \leq +\infty$, by the assumption of the data generation process $y = f(x,t) + \epsilon$, $\epsilon \sim \mathcal{N}(0, \sigma^2)$, and triangular inequality of norm,

$$||\mathbf{y}_z|| \leq ||f(\mathbf{x}_z, \mathbf{t}_z)|| + ||\gamma_z|| \tag{14}$$

$$\leq \sqrt{z}F + ||\gamma_z||, \tag{15}$$

where $\gamma_z$ is a multi-variate Gaussian random variable in $\mathbb{R}^z$ with mean $\mathbf{0}$ and covariance matrix $\sigma^2 \cdot I_z$. Hence $||\gamma_z||/\sigma^2$ is a Chi-squared random variable with degrees of freedom equal to $z$. Then we have with probability at least $1 - \delta/2$,

$$C_1 \leq L_K(zF + 2z\sqrt{\eta_z \sigma^2})/\sigma^2,$$

where $\eta_z = \log(\pi^2 z^2/\delta)$. On the other hand, $C_2$ can be upper bounded as

$$C_2(\tau) \leq \sqrt{2\tau L_K(1 + z \cdot U_K^t/\sigma^2)}.$$

Hence, by choosing $\tau = 1/z^2$, we have

$$(C_1 + L_f)\tau \in \mathcal{O}(1/z),$$

and with probability at least $1 - \delta$, we have

$$\sup_{X \in \mathcal{R}} |f(X,t) - \tilde{f}_n(X,t)| \leq \sqrt{\frac{4L_K + 2U_K/\sigma^2}{z} d\log(1 + z^2 r)} + \sqrt{2d\log(1 + z^2 r) \sup_{x \in \mathcal{R}_n^{1-t}} \sigma_n(x)}$$

$$+ \mathcal{O}(1/z)$$

Therefore, we have that with a probability at least $(1-\delta)^2$ that for both $t = 0$ and $t = 1$

$$\sup_{x \in \mathcal{R}_n^{1-t}} |f(x,t) - \tilde{f}_{\bar{n}_t}(x,t)| \leq \left(\sqrt{\frac{C_K^t}{\bar{n}_t}} + \sqrt{\sup_{x \in \mathcal{R}_n^{1-t}} \sigma_{\bar{n}_t}(x)}\right)\sqrt{d\log\left(\frac{1 + \bar{n}_t^2 r_t}{\delta}\right)} + \mathcal{O}(1/\bar{n}_t),$$

This implies that

$$\sup_{t \in \{0,1\}} \sup_{x \in \mathcal{R}_n^{1-t}} |f(x,t) - \tilde{f}_{\bar{n}_t}(x,t)|$$

$$\leq \sup_{t \in \{0,1\}} \left\{ \left(\sqrt{\frac{C_K^t}{\bar{n}_t}} + \sqrt{\sup_{x \in \mathcal{R}_n^{1-t}} \sigma_{\bar{n}_t}(x)}\right)\sqrt{d\log\left(\frac{1 + \bar{n}_t^2 r_t}{\delta}\right)} + \mathcal{O}(1/\bar{n}_t) \right\}$$

$$\leq \sup_{t \in \{0,1\}} \left\{ \left(\sqrt{\frac{C_K^t}{\bar{n}_t}} + \sqrt{\sup_{x \in \mathcal{R}_n^{1-t}} \sigma_{\bar{n}_t}(x)}\right)\sqrt{d\log\left(\frac{1 + \bar{n}_t^2 r_t}{\delta}\right)} \right\} + \mathcal{O}(1/\bar{n}_0 \wedge \bar{n}_1)$$

$$\leq \sup_{t \in \{0,1\}} \left\{ \left(\sqrt{\frac{C_K^0 \vee C_K^1}{\bar{n}_0 \wedge \bar{n}_1}} + \sqrt{\sup_{x \in \mathcal{R}_n^{1-t}} \sigma_{\bar{n}_t}(x)}\right)\sqrt{d\log\left(\frac{1 + \bar{n}_t^2 r_t}{\delta}\right)} \right\} + \mathcal{O}(1/\bar{n}_0 \wedge \bar{n}_1)$$

$$\leq \sqrt{d}\left(\sqrt{\frac{C_K^0 \vee C_K^1}{\bar{n}_0 \wedge \bar{n}_1}} + \sup_{t \in \{0,1\}}\sqrt{\sup_{x \in \mathcal{R}_n^{1-t}} \sigma_{\bar{n}_t}(x)}\right)\sqrt{\log\left(\frac{1 + (\bar{n}_0 \vee \bar{n}_1)^2 r_t}{\delta}\right)} + \mathcal{O}(1/\bar{n}_0 \wedge \bar{n}_1)$$

By change of variable $(1-\delta)^2 = 1 - \delta'$, we have with probability $1 - \delta'$ for $\delta' \in (0,1)$,

$$\sup_{t \in \{0,1\}} \sup_{x \in \mathcal{R}_n^{1-t}} |f(x,t) - \tilde{f}_{\bar{n}_t}(x,t)|$$

$$\leq \sqrt{d}\left(\sqrt{\frac{C_K^0 \vee C_K^1}{\bar{n}_0 \wedge \bar{n}_1}} + \sup_{t \in \{0,1\}}\sqrt{\sup_{x \in \mathcal{R}_n^{1-t}} \sigma_{\bar{n}_t}(x)}\right)\sqrt{\log\left(\frac{1 + (\bar{n}_0 \vee \bar{n}_1)^2 r_t}{\sqrt{1 - \sqrt{1 - \delta'}}}\right)} + \mathcal{O}(1/\bar{n}_0 \wedge \bar{n}_1)$$

$$= \sqrt{d}\tilde{\mathcal{O}}\left(\sqrt{\frac{C_K^0 \vee C_K^1}{\bar{n}_0 \wedge \bar{n}_1}} + \sqrt{\sup_{x \in \mathcal{R}_n^1} \sigma_{\bar{n}_0}(x) \vee \sup_{x \in \mathcal{R}_n^0} \sigma_{\bar{n}_1}(x)}\right) + \mathcal{O}(1/\bar{n}_0 \wedge \bar{n}_1)$$

$$\square$$

Table 3: $\sqrt{\varepsilon_{\text{PEHE}}}$ across various CATE estimation models with and without COCOA augmentation on Linear and Non-Linear synthetic datasets. Lower $\sqrt{\varepsilon_{\text{PEHE}}}$ corresponds to better performance.

| Model | **Linear** | | **Non-linear** | |
|---|---|---|---|---|
| | **w/o aug.** | **w/ aug.** | **w/o aug.** | **w/ aug.** |
| TARNet | $0.93_{\pm.09}$ | $0.81_{\pm.02}$ | $7.41_{\pm.23}$ | $6.64_{\pm.11}$ |
| CFR-Wass | $0.87_{\pm.05}$ | $0.74_{\pm.05}$ | $7.32_{\pm.21}$ | $6.22_{\pm.07}$ |
| CFR-MMD | $0.91_{\pm.04}$ | $0.78_{\pm.06}$ | $7.35_{\pm.19}$ | $6.28_{\pm.10}$ |
| T-Learner | $0.90_{\pm.01}$ | $0.89_{\pm.01}$ | $7.68_{\pm.12}$ | $7.51_{\pm.07}$ |
| S-Learner | $0.64_{\pm.01}$ | $0.63_{\pm.01}$ | $7.22_{\pm.01}$ | $6.92_{\pm.01}$ |
| BART | $0.65_{\pm.00}$ | $0.30_{\pm.00}$ | $5.49_{\pm.00}$ | $4.50_{\pm.00}$ |
| CF | $0.63_{\pm.00}$ | $0.27_{\pm.00}$ | $5.46_{\pm.00}$ | $4.46_{\pm.00}$ |

## D  ADDITIONAL EMPIRICAL RESULTS

In this section, we present additional results for the completeness of the empirical study for COCOA. Specifically, we (*i*) add the results for the synthetic datasets, (*ii*) provide details for the toy example used to generate Figure 1b, (*ii*) present more visualizations illustrating the effect of contrastive learning, (*iv*) study the performance of our proposed method on ATE estimation, (*v*) conduct ablation studies on the local regression module, (*vi*) present additional results to demonstrate robustness against overfitting, and (*vii*) perform ablation studies on different parameters for the contrastive learning module.

### D.1  RESULTS FOR SYNTHETIC DATA

In this section, we present the $\sqrt{\varepsilon_{\text{PEHE}}}$ results for various CATE estimation models on synthetic datasets, both linear and non-linear. Table 3 summarizes the performance of each model with COCOA augmentation (w/ aug.) and without augmentation (w/o aug.). Lower $\sqrt{\varepsilon_{\text{PEHE}}}$ indicates better performance. The results demonstrate that COCOA augmentation consistently improves the performance across different models and datasets.

### D.2  TRADE-OFF TOY EXAMPLE

In this section, we synthetically generate a dataset for a binary treatment scenario with 1000 samples per treatment group and $d = 4$ features. We sample a vector of coefficients,

$$\beta \sim \mathcal{N}(\mathbf{0}, \mathbf{I}_d)$$

where $\mathbf{0} \in \mathbb{R}^d$ is the zero vector and $\mathbf{I}_d$ is the $d \times d$ identity matrix.

Next, we generate feature vectors $X \in \mathbb{R}^d$ for the two treatment groups:

$$X_0 \sim \mathcal{N}(-\mathbf{1}, 0.5\mathbf{I}_d)$$

and,

$$X_1 \sim \mathcal{N}(\mathbf{1}, 0.5\mathbf{I}_d)$$

where $-\mathbf{1} \in \mathbb{R}^d$ and $\mathbf{1} \in \mathbb{R}^d$ are vectors with all elements equal to -1 and 1, respectively, and $\mathbf{I}_d$ is the $d \times d$ identity matrix.

The potential outcomes are generated as follows:

$$Y_0 = (\beta^T X_0)^3 + \mathcal{N}(0, 0.1)$$

and

$$Y_1 = (\beta^T X_1)^2 + \mathcal{N}(0, 0.1)$$

We implement a function to augment the datasets using a nearest-neighbor approach with a specified radius (radius is set to 8). The augmentation involves imputing potential outcomes for individuals from the opposite treatment group if they have at least three close neighbors within the specified radius. We then perform linear regression to impute the outcomes. We include further empirical results in Figure 4.

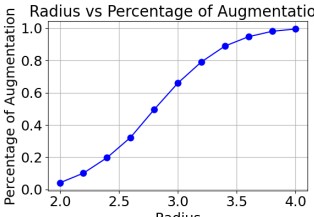 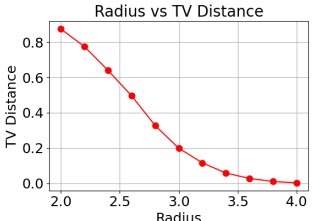 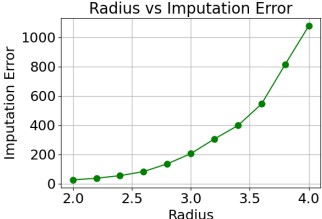

Figure 4: Trade-off between imputation error and statistical disparity.The first plot displays the percentage of augmentation as a function of the radius. The second and third plots show the Total Variation (TV) distance and imputation error, respectively, for different radius values.

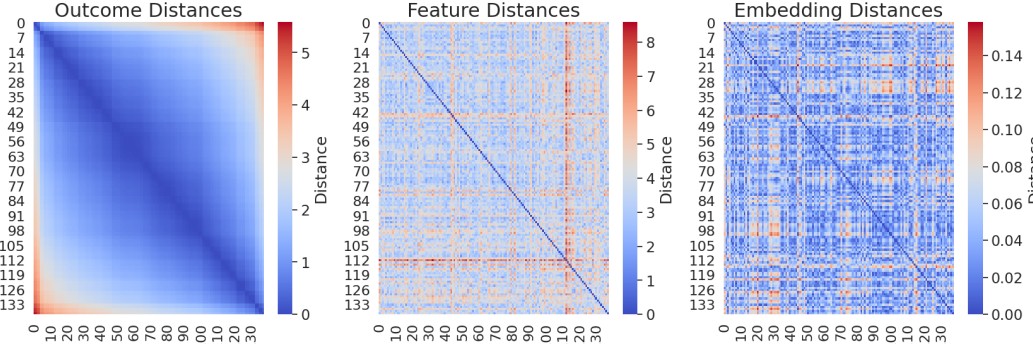

Figure 5: Comparison between euclidean distance and latent distance lerned by contrastive learning for the IHDP dataset (treatment group). The first heatmap illustrates the outcome distances. The second heatmap shows the feature distances, reflecting differences between feature vectors. The third heatmap presents the embedding distances, demonstrating how the learned embeddings capture the same similarities as the potential outcome.

### D.3 CONTRASTIVE LEARNING MOTIVATION

In this section, we provide more motivation for the use of contrastive learning to learn a representation space in which we identify similar individuals instead of using traditional methods (e.g., euclidean distance the ambient space). Figures 5 and 6 illustrate this. We also include an ablation on the effect of the embedding dimension for contrastive learning on the learned representation for the IHDP dataset as illustrated in Figure 7.

### D.4 ATE ESTIMATION PERFORMANCE

In this section, we provide additional empirical results when applying our methods to ATE estimation. The Average Treatment Effect (ATE) is defined as:

$$\tau_{\text{ATE}} = \mathbb{E}[Y_1 - Y_0].$$

The error of ATE estimation is defined as:

$$\varepsilon_{\text{ATE}} = \left| \hat{\tau}_{\text{ATE}} - \tau_{\text{ATE}} \right|, \tag{16}$$

Our results are summarized in Tables 4, 5, and 6. We observe that our methods, while not tailored for ATE estimation, still bring some benefits for a subset of the estimation models.

### D.5 LOCAL REGRESSION MODULE

In this section, we compare the performance of using Gaussian Processes (GP)with different kernels vs. local linear regression. We next define the local linear regression module and present the empirical results in Table 7.

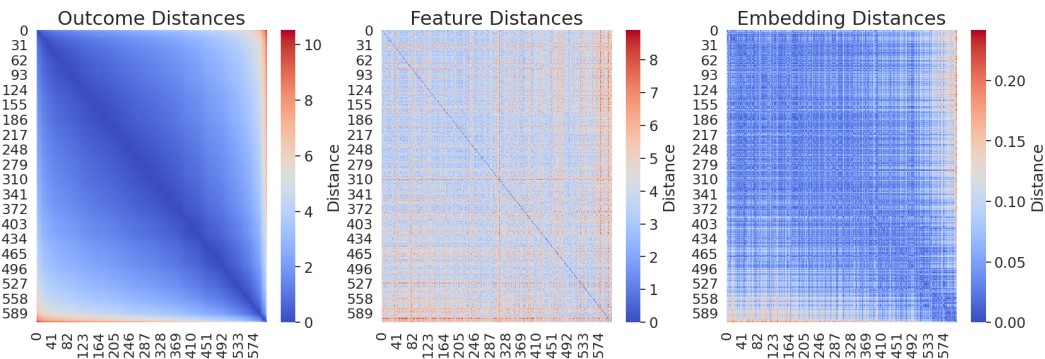

Figure 6: Comparison between euclidean distance and latent distance lerned by contrastive learning for the IHDP dataset (control group). The first heatmap illustrates the outcome distances. The second heatmap shows the feature distances, reflecting differences between feature vectors. The third heatmap presents the embedding distances, demonstrating how the learned embeddings capture the same similarities as the potential outcome.

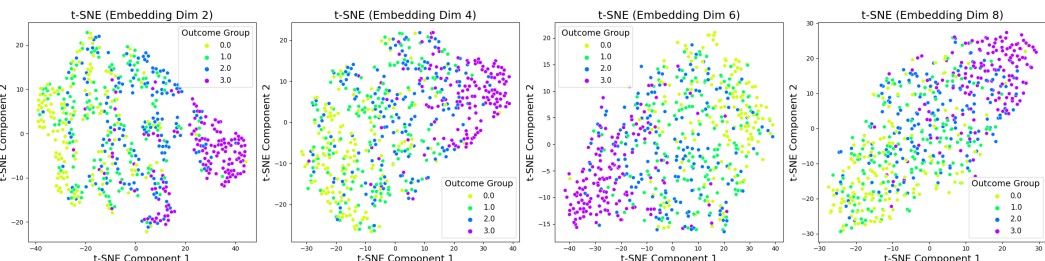

Figure 7: t-SNE visualizations of the IHDP dataset control group embeddings for different embedding dimensions. The figure illustrates t-SNE plots for the control group with embedding dimensions of 2, 4, 6, and 8. The points are colored based on outcome groups, created by dividing the outcomes into four quantiles. Each subplot shows how the embeddings distribute in a 2D space, capturing the relationship between the learned embeddings and outcome groups. Outcome groups represent different quantile ranges of potential outcomes: Group 0 (yellow) includes the lowest quantile, Group 1 (cyan) includes the second lowest, Group 2 (blue) includes the second highest, and Group 3 (magenta) includes the highest quantile.

Table 4: $\varepsilon_{\text{ATE}}$ across various CATE estimation models, with COCOA augmentation (w/ aug.) and without augmentation (w/o aug.) in Twins, Linear, and Non-Linear datasets. Lower $\varepsilon_{\text{ATE}}$ corresponds to the better performance.

| | TWINS | | LINEAR | | NON-LINEAR | |
|---|---|---|---|---|---|---|
| MODEL | W/O AUG. | W/ AUG. | W/O AUG. | W/ AUG. | W/O AUG. | W/ AUG. |
| TARNET | $0.33_{\pm.19}$ | $0.41_{\pm.29}$ | $0.10_{\pm.02}$ | $0.04_{\pm.02}$ | $0.23_{\pm.13}$ | $0.04_{\pm.02}$ |
| CFR-WASS | $0.47_{\pm.16}$ | $0.14_{\pm.09}$ | $0.13_{\pm.04}$ | $0.06_{\pm.01}$ | $0.19_{\pm.09}$ | $0.03_{\pm.01}$ |
| CFR-MMD | $0.19_{\pm.09}$ | $0.18_{\pm.12}$ | $0.12_{\pm.05}$ | $0.05_{\pm.03}$ | $0.25_{\pm.15}$ | $0.04_{\pm.01}$ |
| T-LEARNER | $0.02_{\pm.02}$ | $0.05_{\pm.03}$ | $0.01_{\pm.01}$ | $0.01_{\pm.01}$ | $0.05_{\pm0.02}$ | $0.05_{\pm.01}$ |
| S-LEARNER | $0.89_{\pm.03}$ | $0.79_{\pm.07}$ | $0.03_{\pm.01}$ | $0.05_{\pm.01}$ | $0.45_{\pm.05}$ | $0.27_{\pm.02}$ |
| BART | $0.28_{\pm.08}$ | $0.21_{\pm.10}$ | $0.37_{\pm.00}$ | $0.07_{\pm.01}$ | $0.80_{\pm.00}$ | $0.26_{\pm.00}$ |
| CF | $0.28_{\pm.06}$ | $0.14_{\pm.15}$ | $0.39_{\pm.00}$ | $0.06_{\pm.01}$ | $0.77_{\pm.00}$ | $0.32_{\pm.00}$ |

Table 5: $\varepsilon_{\text{ATE}}$ across various CATE estimation models, with COCOA augmentation (w/ aug.), without augmentation (w/o aug.), and with Perfect Match augmentation in News and IHDP datasets. Lower $\varepsilon_{\text{ATE}}$ corresponds to the better performance.

| | NEWS | | IHDP | |
|---|---|---|---|---|
| MODEL | W/O AUG. | W/ AUG. | W/O AUG. | W/ AUG. |
| TARNET | $0.97_{\pm.45}$ | $0.96_{\pm.38}$ | $0.12_{\pm.05}$ | $0.07_{\pm.03}$ |
| CFR-WASS | $1.00_{\pm.29}$ | $0.75_{\pm.22}$ | $0.10_{\pm.03}$ | $0.05_{\pm.02}$ |
| CFR-MMD | $0.89_{\pm.38}$ | $0.71_{\pm.22}$ | $0.16_{\pm.04}$ | $0.09_{\pm.04}$ |
| T-LEARNER (NN) | $0.49_{\pm.26}$ | $0.76_{\pm.20}$ | $0.27_{\pm.06}$ | $0.07_{\pm.03}$ |
| S-LEARNER (NN) | $0.40_{\pm.06}$ | $0.49_{\pm.27}$ | $1.72_{\pm.21}$ | $0.40_{\pm.02}$ |
| BART | $0.77_{\pm.13}$ | $0.60_{\pm.00}$ | $0.02_{\pm.01}$ | $0.02_{\pm.01}$ |
| CAUSAL FORESTS | $0.72_{\pm.01}$ | $0.60_{\pm.00}$ | $0.11_{\pm.01}$ | $0.03_{\pm.02}$ |
| PERFECT MATCH | $2.00_{\pm1.01}$ | | $0.24_{\pm.20}$ | |

**Local Linear Regression.** For a fixed individual $x$ who received treatment $t$, and has a selected neighbors $D_{x,t}$. Under the assumption that we can locally approximate the true function with a linear function. Suppose $X_D$ is the matrix of the observed feature values in $D_{x,t}$ augmented with a column of ones for the intercept, and $Y_D$ is the column vector of observed factual outcomes. The local linear regression coefficients, $\hat{\beta}$, are computed as:

$$\hat{\beta} = (X_D^T X_D)^{-1} X_D^T Y_D$$

Then we impute the value of $x$ as $\hat{y} = [1, x]^T \hat{\beta}$.

### D.6 Ablation For Contrastive Learning Parameters

In this section, we provide a comprehensive set of ablation studies for the effect of the hyper-parameters of the contrastive learning module.

**Ablation on K and R.** We provide extra ablation studies on the IHDP dataset and the Non-linear dataset to study the effect of *(i)* the number of neighbors (K) and *(ii)* the embedding radius (R) on both $\varepsilon_{PEHE}$ and $\varepsilon_{ATE}$. We observe a consistently enhanced performance across different CATE estimation models. See results in figures 10 and 11. We also provide ablation studies on the sensitivity of the proposed Contrative Learning module to the parameter $\epsilon$, which is used to create the training points for the contrastive learning module by creating positive and a negative dataset, see Section 5 for more details.

**Ablation on the sensitivity parameter** $\epsilon$ We provide ablation on the sensitivity parameter $\epsilon$, a similarity classifier for the potential outcomes (see Section 5 for a detailed description). The results for the $\varepsilon_{PEHE}$ as a function of $\epsilon$ are presented in Figure 8. It can be observed that the error of CATE estimation models is consistent for a wide range of $\epsilon$, demonstrating the robustness of COCOA to the choice of hyper-parameters.

Table 6: $\varepsilon_{\text{ATE}}$ across different similarity measures: Contrastive Learning (CL), propensity scores (PS), and Euclidean distance (ED), using CFR-Wass across IHDP, News, and Twins datasets.

| MEASURE OF SIMILARITY | ED | PS | CL |
|---|---|---|---|
| IHDP | $3.12_{\pm 1.33}$ | $3.85_{\pm .22}$ | $\mathbf{0.05}_{\pm .02}$ |
| NEWS | $0.68_{\pm .20}$ | $\mathbf{0.54}_{\pm .25}$ | $0.75_{\pm .22}$ |
| TWINS | $\mathbf{0.13}_{\pm .15}$ | $0.46_{\pm .09}$ | $0.14_{\pm .09}$ |

Table 7: Comparison of $\varepsilon_{\text{PEHE}}$ and $\varepsilon_{\text{ATE}}$ across different local regression modules: Gaussian Process (GP) with various kernels (DotProduct, RBF, and Matern) and Linear Regression. The first three rows present $\sqrt{\varepsilon_{\text{PEHE}}}$, while the subsequent three rows display $\varepsilon_{\text{ATE}}$.

| LR | GP (DOTPRODUCT) | GP (RBF) | GP (MATERN) | LINEAR REGRESSION |
|---|---|---|---|---|
| IHDP | $\mathbf{0.63}_{\pm .01}$ | $0.63_{\pm .00}$ | $0.65_{\pm .02}$ | $0.75_{\pm .01}$ |
| NEWS | $3.56_{\pm .01}$ | $3.55_{\pm .04}$ | $\mathbf{3.44}_{\pm .05}$ | $3.53_{\pm .08}$ |
| TWINS | $\mathbf{0.51}_{\pm .11}$ | $0.51_{\pm .02}$ | $0.54_{\pm .04}$ | $0.68_{\pm .08}$ |
| IHDP | $0.02_{\pm .01}$ | $\mathbf{0.01}_{\pm .00}$ | $0.03_{\pm .01}$ | $0.09_{\pm .01}$ |
| NEWS | $0.60_{\pm .00}$ | $0.24_{\pm .12}$ | $\mathbf{0.05}_{\pm .03}$ | $0.21_{\pm .10}$ |
| TWINS | $\mathbf{0.21}_{\pm .10}$ | $0.24_{\pm .04}$ | $0.29_{\pm .04}$ | $0.38_{\pm .10}$ |

### D.7 OVERFITTING TO THE FACTUAL DISTRIBUTION

In this section, we provide more empirical results on the robustness against overfitting to the factual distribution for the Linear and Non-Linear synthetic datasets, as presented in Figure 9.

## E LIMITATIONS

It is important to note that when the statistical disparity between the treatment groups is zero, the counterfactual data augmentation method will likely not bring any benefits. Similarly, when there is a total discrepancy between the two groups (i.e., disjoint supports), no benefits will be observed. Moreover, as the fundamental problem of causal inference implies that CATE values are unobservable, it is challenging to fine-tune the parameters of COCOA.

## F COMPUTATIONAL RESOURCES

The experiments in this paper are not computationally expensive to conduct and were performed on the following GPU: NVIDIA GeForce RTX 3090.

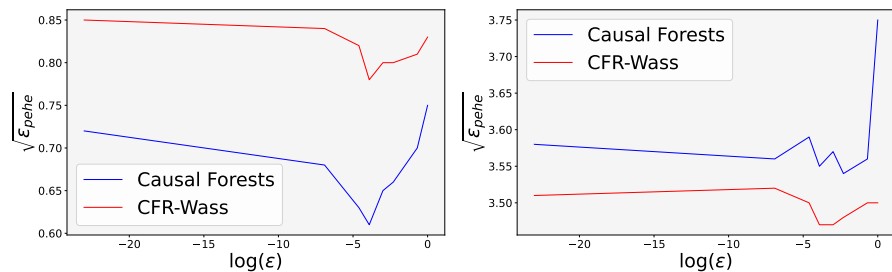

Figure 8: $\varepsilon_{\text{PEHE}}$ as a function of the similarity sensitivity parameter $\epsilon$. The figure on the left presents results for the IHDP dataset, while the one on the right is for the News dataset. Performances of two different models (CFR-Wass and Causal Forests) are plotted for both datasets.

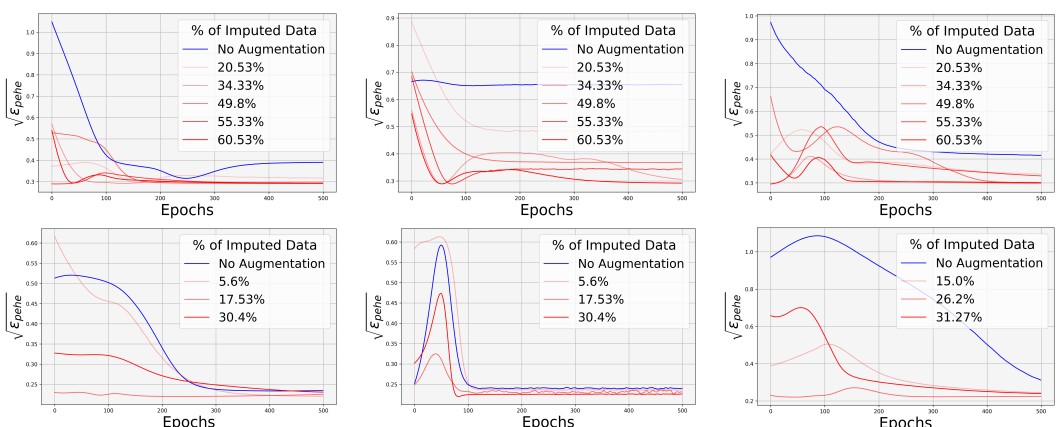

Figure 9: Comparison of training progression for TARNet, CFR-Wass, and T-learner models on linear and non-linear datasets. Top row: Models trained on the linear dataset, showcasing TARNet, CFR-Wass, and T-learner, respectively. Bottom row: The same models trained on the non-linear dataset. This visualization demonstrates the effects of COCOA on preventing overfitting across different data complexities and the performance of three CATE estimation models trained with various levels of data augmentation.

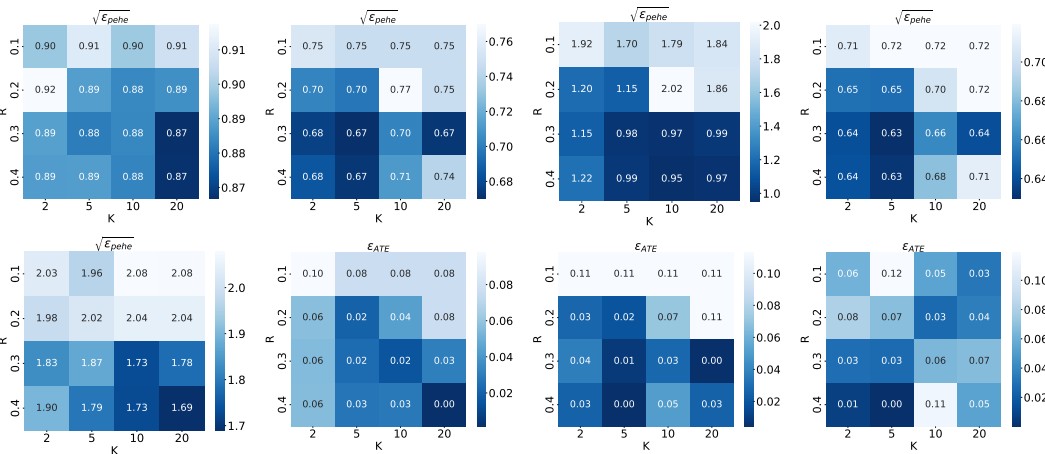

Figure 10: Ablation studies on the impact of the size of the $\epsilon-$Ball (R) and the number of neighbors (K) on the performance. The first row from left to right: IHDP with TARNet, BART, S-Learner, and Causal Forests. The second row: IHDP with Causal Forests, T-Learner, BART, and TARNet. These studies illustrate the trade-off between minimizing the discrepancy between the distributions—achieved by reducing K and increasing R—and the quality of the imputed data points, which is achieved by decreasing R and increasing K.

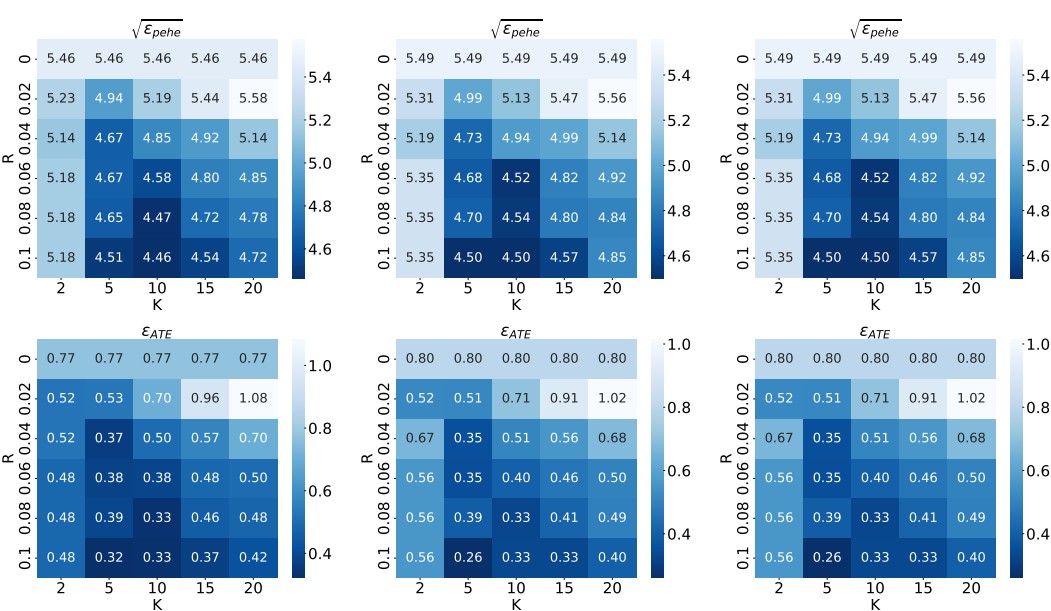

Figure 11: Ablation studies on the Non-linear dataset. Top row from left to right: Causal Forests (PEHE), BART (PEHE), TARNet (PEHE). Bottom row from left to right: Causal Forests (ATE), BART (ATE), TARNet (ATE). Each pair of images represents the performance of the respective models evaluated in terms of Precision in Estimation of Heterogeneous Effect (PEHE) and the error in Average Treatment Effect (ATE) estimation on a non-linear dataset.

