# OpenReview forum: "Potential Outcome Imputation for CATE Estimation"
_ICLR.cc/2025/Conference — Submitted to ICLR 2025_

### Official Review · Reviewer_aKbR · 2024-10-17

**Soundness:** 2
**Presentation:** 2
**Contribution:** 2
**Rating:** 3
**Confidence:** 4

**Summary:**

The paper aims to enhance the performance of CATE estimation methods through potential outcome imputation in a pre-processing step. Specifically,
the paper introduces a contrastive learning approach to reliably impute counterfactual outcomes for subsets of individuals for which a sufficient number
of neighbors in a latent space exist in the factual dataset.

**Strengths:**

- The paper introduces an interesting idea for data augmentation for CATE estimation.
- The paper proposes a pre-processing method that can be combined with any CATE estimator.

**Weaknesses:**

- Limited referencing to support claims.
- Proofs are not stated or referenced in the main papers.
- The method only applies to data with continuous outcomes. This is not clearly communicated in the beginning.
It is unclear, how the method would need to be extended to categorical/discrete outcomes. Especially for binary outcomes, such as survival, I am not sure how the method could work.
- Concern on underlying reasoning/intuition of the method: The paper reasons that individuals with similar outcomes should be treated as similar. Why would this make sense if the
overall task is to estimate the treatment effect and not the potential outcome?
- The method extensively relies on the choice of certain hyperparameters. It is unclear how one would chose those parameters in practice.
Without such knowledge, the contribution of the paper for real-world applications is limited.
- The line of thought in the paper is difficult to follow in its current representation. For example, notation and formulas are referenced before they are
introduced, motivation for theoretical theorems and assumptions is lacking.
- Theoretical insights: Although the paper states much mathematical theory, it does not become clear why COCOA enhances CATE performance in general.
Section 6.1 only states that asymptotically, COCOA converges to an RCT. However, asymptotically, i.e., with sufficient data samples, we would not even need
an imputation method. Section 6.2 only provides finite sample guarantees for GPs (under additional assumptions) and does not proof the applicability of COCOA
in general.
- See questions.

**Questions:**

- Definition 3.3: Why is this loss function reasonable? Where does it stem from?
- Why does the smoothness property ensure reliable imputation? This needs to be explained/justified further.
- Training the classifier g: Especially in high-dimensional settings, most samples will be dissimilar. In the "worst-case", this will lead to a too small (or even an empty) similarity set,
such that no sample will be perturbed (as the authors also state in line 255). How does the dimensionality affect the training of g?
- How would one know how to choose the parameter k?
- The paper proposes to extend the factual dataset by potential outcome imputation of certain samples. From my understanding, this happens in an iterative fashion,
i.e., the PO for sample i is estimated and added to the dataset. For estimating the next PO for sample j, the new dataset is employed. How does the estimation error
propagate and affect the later estimates?
- Equations 4 needs more explanation: What do 0 and 1 stand for? What is the purpose of g? I am aware that there is some explanation later on.
However, the presentation of Section 5 could be strongly improved to facilitate the reader's understanding.
- Why is it reasonable to assume Lipschitzness of the potential outcome function? IMHO, this is quite restrictive for general covariate spaces.
- Lines 422/423: "this error" needs more explanation. How exactly does Theorem 6.5 solve the issue?
- The paper established that the method provides assymptotically unbiased estimates. This fact does not differentiate the paper from other methods.
It would be interesting to compare the convergence rates to draw more educated conclusions about the usefulness of the method with respect to asymptotic unbiasedness.
- Table 1: What is the explanation for COCOA not enhancing the average performance or significantly increasing the variance across runs for some CATE estimators and dataset combinations?
Is this caused by no or only a few augmented samples?
- Figure 3: How can the much slower convergence of COCOA for increasing % of imputed data be explained?
- ATE estimation: How can the mixed performance of COCOA for ATE estimation be explained (appendix D.4)
- Minor language errors.

---

### Official Review · Reviewer_F9mV · 2024-11-03

**Soundness:** 2
**Presentation:** 3
**Contribution:** 2
**Rating:** 3
**Confidence:** 5

**Summary:**

The authors propose a data augmentation strategy to improve the accuracy of CATE estimation. To address the “missing data” problem in causal inference (we only observe an individual’s outcome under treatment or control), the authors propose to use data augmentation to impute the missing counterfactual outcomes and then estimate the CATE function on the augmented dataset.

**Strengths:**

The paper studies the problem of CATE estimation, which is a fundamental problem in causal inference, and seeks to employ modern machine learning strategies such as data augmentation and contrastive learning for addressing this problem.

The ideas of applying data augmentation and contrastive learning have not yet been applied in the causal inference literature.

**Weaknesses:**

The paper does not discuss the R-learner [Nie & Wager 2021], which is a widely used method for heterogeneous treatment effect estimation in observational studies. Nie & Wager, 2021 first use machine learning methods to estimate the treatment propensity function and the conditional mean outcome functions. After that, they optimize a data-adaptive objective function that targets the conditional average treatment effect. Nie & Wager 2021 demonstrate that if the conditional mean outcomes and the treatment propensity can be estimated (at a certain rate), it is possible to obtain fast rates of convergence for estimating the CATE function. It is not clear that the authors' proposed approach offers any benefit if one already is using the R-learner, which Kennedy et al. [2024] show is minimax for estimating CATEs under a set of smoothness assumptions.

Clarity issue: The authors write that one of the main challenges in CATE estimation is “statistical discrepancy between distinct treatment groups” – It is not clear whether the authors are referring to the fact that treatment and control groups may differ along observables (covariates) or unobservables (or both). The first paragraph of the introduction suggests that this problem disappears in RCTs but appears when running observational studies, so it seems as though the authors are referring to the fact that observational studies may suffer from unmeasured confounding. However, later in the paper, the authors describe that they assume conditional unconfoundedness, so it seems that by “statistical discrepancy between treatment and control groups” the authors are referring to differences between observables between treatment and control groups. In RCTs, we can assume that there’s no statistical difference in observables between treatment and control groups in the infinite-sample population regime, but it is important to emphasize that a statistical difference between treatment and control groups may persist in finite-sample RCTs.

Nie, Xinkun, and Stefan Wager. "Quasi-oracle estimation of heterogeneous treatment effects." Biometrika 108.2 (2021): 299-319.

Edward H Kennedy, Sivaraman Balakrishnan, James M Robins, and Larry
Wasserman. Minimax rates for heterogeneous causal effect estimation. The
Annals of Statistics, 52(2):793–816, 2024.

**Questions:**

It is not clear to me whether applying this data augmentation approach is preferable to simply fitting the T-Learner? How does the authors’ proposal compare to an approach where we estimate the conditional mean function for the treated and control units separately, then estimate the CATE by subtracting these two conditional mean functions (T-Learner)?

---

### Official Review · Reviewer_nW9p · 2024-11-04

**Soundness:** 2
**Presentation:** 1
**Contribution:** 2
**Rating:** 3
**Confidence:** 4

**Summary:**

This paper proposes a method to impute potential outcomes for the estimation of treatment effects. The method can be processed in two steps: (1) identify a subset of individuals for which counterfactual outcomes can be estimated reliably, (2) use contrastive learning to impute missing potential outcomes for the selected individuals and augment the original data with the imputed values. It is nice that the paper also studies the theoretical properties of the proposed method, but the method itself is not novel. The issues raised in this paper are common and known to everyone in the fields.

**Strengths:**

The paper studies a classical problem.

**Weaknesses:**

The novelty and presentation of the paper can be improved.

**Questions:**

1. The paper is not well-motivated. The motivation is a well-known fact that the worse the covariate distribution in the original dataset, the harder it is to estimate treatment effects. A motivation from the drawbacks of the other methods in the literature can be more convincing.
2. Subset identification: Is the identified subset the units with >=k near neighbors? Please clarify. The authors ignore the literature on identifying domains that overlap well across treatment groups. How does the proposed method compare with matching methods?
3. Similarity learning: The authors should put some effort into discussing why learning a representation space is better than using the raw data. Why use contrastive learning? Why is it better than other deep learning methods? How to choose the sensitivity parameter epsilon and the threshold k?
4. Local imputation: The authors should clearly state the smoothness conditions. What is a sufficient number of close neighbors (I guess it is >=k in the paper, but the authors should be clear about that)?
5. CATE estimation: the authors mention the proposed method avoids overfitting several times but need to justify why this is the case
6. The term “hypothesis” is confusing to the broader audience. It can be confused with hypothesis testing in statistics.

---

### Official Review · Reviewer_tg1V · 2024-11-04

**Soundness:** 3
**Presentation:** 4
**Contribution:** 2
**Rating:** 5
**Confidence:** 5

**Summary:**

This paper proposes a model-agnostic data augmentation method to enhance CATE by imputing missing counterfactual outcomes for a targeted subset of individuals. First, they use uses contrastive learning to identify individuals for whom counterfactual outcomes should be imputed. Then, they use Gaussian Processes to apply a local regression module to estimate these outcomes. A CATE estimation method is then used on this augmented dataset. The authors provide empirical evidence demonstrating that this data augmentation improves standard CATE estimators, alongside theoretical results in both population and finite sample setting supporting the method.

**Strengths:**

- The paper is well-structured and thorough. The flow is lucid.

-  The proposed idea is straightforward and comes with theoretical guarantees

- The idea of selecting a subset of individuals for whom augmentation is reliable is interesting.

- I like the way the experiments are presented, where they compare prior approaches with and without their augmented samples.

**Weaknesses:**

- The design choice of $g_\theta$ is unclear. One could perhaps come up with cases where points $x$ and $x'$ have small differences in outcomes under treatment 1 (i.e., $ y_1 - y_1'$ is small) but large differences under treatment 0 (i.e., $ y_0 - y_0'$). This scenario would challenge the model's assumptions.

- Given the importance of $g_\theta$, the selection of $\epsilon$ that influences $g_\theta$ training also becomes crucial. While it may appear that $\epsilon$ has little impact in these synthetic datasets, it could be quite significant in high-dimensional datasets commonly encountered in practice. Do the authors have any insights on how to determine this parameter more generally?

- The experimental section lacks depth, as it primarily compares against very old baselines rather than recent CATE approaches. For example, see the methods listed in CATENets: [https://github.com/AliciaCurth/CATENets](https://github.com/AliciaCurth/CATENets).

- Moreover, the paper doesn’t compare against other data augmentation methods. I have read many CATE papers that used techniques like Gaussian Processes, nearest neighbors, IPW-based imputations, and GANs for augmentation. Including these comparisons would better contextualize the merits of this work.

- Ablation analysis that considers COCOA with and without $g_\theta$ is missing in the paper. I feel that this is an important ablation.

**Questions:**

The experiments are not comprehensive. More recent baselines and explicit comparison with data augmentation based methods explored for CATE in literature are required.

---

### Meta-Review · Area_Chair_iJhH · 2024-12-20

**Metareview:**

The paper introduces a model-agnostic data augmentation method for Conditional Average Treatment Effect (CATE) estimation. Using contrastive learning and Gaussian Processes, it imputes missing potential outcomes, reducing statistical discrepancies between treatment groups while minimizing imputation errors. Key contributions include theoretical guarantees, improved accuracy and robustness for various CATE models, and empirical validation across benchmark datasets.

The reviewers have pointed the lack of the importance of the issue, comparative experiments with previous studies, and in-depth discussions based on them. It was also noted that there is confusion in the use of terms in the discussion. The authors did not refute this.

**Additional Comments On Reviewer Discussion:**

Half of the reviewers pointed out the inadequacy of the comparisons, especially in the experimental portion of the study. The other half of the reviewers correctly pointed out the lack of importance of the issues presented in the paper and the confusion in the terminology and design philosophy used. All reviewers shared the same negative view of this paper.

---

### Decision · Program_Chairs · 2025-01-22

Reject